# Trust within human-machine collectives depends on the perceived consensus about cooperative norms

Kinga Makovi [1], Anahit Sargsyan[2], Wendi Li[3], Jean-François Bonnefon [4] & Talal Rahwan [3] ✉

With the progress of artificial intelligence and the emergence of global online communities, humans and machines are increasingly participating in mixed collectives in which they can help or hinder each other. Human societies have had thousands of years to consolidate the social norms that promote cooperation; but mixed collectives often struggle to articulate the norms which hold when humans coexist with machines. In five studies involving 7917 individuals, we document the way people treat machines differently than humans in a stylized society of beneficiaries, helpers, punishers, and trustors. We show that a different amount of trust is gained by helpers and punishers when they follow norms over not doing so. We also demonstrate that the trust-gain of norm-followers is associated with trustors' assessment about the consensual nature of cooperative norms over helping and punishing. Lastly, we establish that, under certain conditions, informing trustors about the norm-consensus over helping tends to decrease the differential treatment of both machines and people interacting with them. These results allow us to anticipate how humans may develop cooperative norms for human-machine collectives, specifically, by relying on already extant norms in human-only groups. We also demonstrate that this evolution may be accelerated by making people aware of their emerging consensus.

From humble chatbots to state-of-the-art Artificial Intelligence (AI), intelligent machines are increasingly participating in mixed human-bot collectives[1-3]. These new collectives will face similar challenges of cooperation, exploitation, and norm stabilization that human societies have gone through and continue to struggle with[4-10]. Many online communities already offer a glimpse into these future human-bot collectives. Wikipedia, Twitter, Reddit, YouTube, Twitch and Discord are all examples of communities populated by humans and bots who can help or hinder each other. In all these online communities, bots can create content themselves, promote, suppress or denigrate human-created content, and even act against human users by issuing warnings, muting or deafening their accounts, reporting them to moderators, or outright banning them[11-14]. In turn, humans can promote or suppress bot-created content, but they can also take action against bots, for example by reporting them to moderators. Human-bot collectives often struggle when it comes to defining the norms that regulate human-bot interactions both in the lab[15-20] and in the field, as illustrated by the difficult task of Wikipedia and Twitter moderators when it comes to explaining what constitutes inappropriate behavior from bots[21,22].

Human-bot collectives can take many forms, and afford many variants of basic interactions such as helping or hindering. In this

[1]Social Science Division, New York University Abu Dhabi, Abu Dhabi, UAE. [2]School of Social Sciences and Technology, Technical University of Munich, Munich, Germany. [3]Computer Science, Science Division, New York University Abu Dhabi, Abu Dhabi, UAE. [4]Toulouse School of Economics and Quantitative Social Sciences, CNRS (TSM-R), Toulouse, France. ✉e-mail: talal.rahwan@nyu.edu

article, we report a series of online experiments examining how humans behave in a mixed collective where humans and bots can help or hinder each other, how their behavior correlates with the norms they perceive to hold in this new landscape, and how their behavior changes when they are informed of the consensus that is emerging about these norms. We use a stylized society (closely following the design in ref. 23) where agents can be Beneficiaries, Helpers, Punishers, and Trustors, and where decisions have real financial consequences for the people constituting these stylized societies. We therefore follow the tradition of using financial incentives as an instrument to capture the many forms of cooperation that can take place in real life, offline or online[24]. Beneficiaries, Helpers and Punishers interact through a third-party punishment game, where Helpers decide whether to share resources with Beneficiaries, and Punishers decide whether to punish, at a cost, the Helpers who did not share. Trustors interact with Helpers or Punishers through a trust game, where they decide whether to invest resources in Helpers or Punishers, after being informed of their behavior in the third-party punishment game. An illustration of this stylized society is depicted in Fig. 1.

Unlike in the experiment by Jordan and colleagues[23], Beneficiaries, Helpers and Punishers in our experiments can be played by either bots or humans. Introducing bots in our stylized society raises a methodological and a theoretical question. From a methodological perspective, the issue at stake is that bots, unlike humans, do not care about money. As a result, one may question our choice to use economic games as a proxy for real-life interactions between bots and humans. To justify our paradigm, we show that human participants believe that bot participants behave as if they had preferences, be it for money (the currency in our experiment), or for real-life currencies like collecting likes on social media, or avoiding sanction and bans in online communities such as Wikipedia. This is what we accomplish in Study 1. More precisely, we demonstrate that while human participants do not believe that bots 'want' to earn money, collect likes, or avoid bans (in the sense that they feel a need or desire for these outcomes), they do believe that the bots behave as if they had all these preferences, because of the way they are programmed. This being established, the theoretical question is

why people may share resources with bots, money or otherwise. While altruism is an unlikely explanation[25], confusion is always an option[26], although we take extensive precautions so that participants understand the incentive structure of our stylized society. The explanation we focus on, instead, is signaling: humans help bots in order to signal to other humans that they are trustworthy.

Here we show that actors earn trust by sharing and by punishing those who do not share, but less so when they share with bots or punish bots. As a result, trust is not as easily established in mixed human-bot collectives. We also provide correlational, experimental, and qualitative evidence that, under certain conditions, the awareness that a majority of participants believe one should help (humans or bots) tends to decrease the differential treatment of bots and humans. These data allow us to anticipate how humans might develop cooperative norms for human-machine collectives, and how this evolution could be accelerated by making humans aware of their emerging consensus.

## Results

The five studies we conduct build on one another sequentially. In Study 1, we show that our stylized society is a proxy for real-life interactions between bots and humans, since participants believe that bots behave as if they had preferences, for money, for collecting likes on Twitter, or for avoiding bans on Wikipedia. In Study 2, we document the differences in the way humans treat bots vs. humans, as well as the difference in the way they treat other humans who interact with bots vs. others who interact with humans in the third-party punishment game. We also establish differences in trust-gain resulting from helping and punishing, based on the identity of the actors involved. In Study 3, we provide evidence that the perceived consensus around the norms of sharing and punishing correlates with the trust that norm-followers gain over non-followers, across each of the different configurations that we examine in the third-party punishment game. Study 4 and Study 5, which follow a within-person and a between-person design, respectively, provide causal, and complementary evidence that clarifying the norm-consensus may amplify the trust-gain of norm-followers in conditions with bot participants.

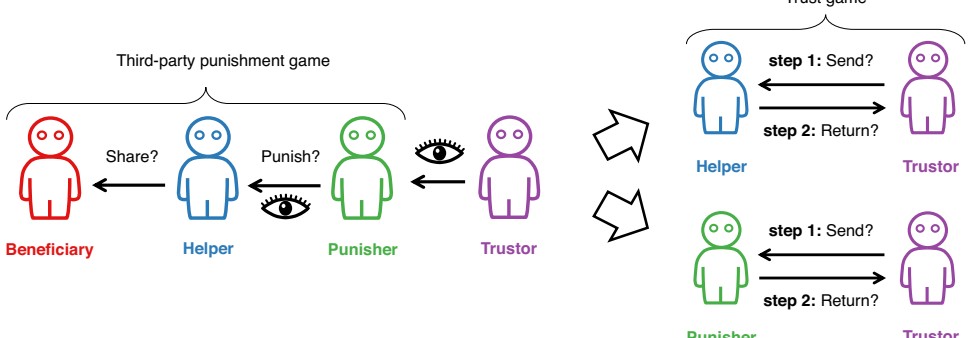

**Fig. 1 | Third-party punishment game followed by a trust game.** First, three players participate in a third-party punishment game. In this game, Helpers and Punishers are initially given resources, while Beneficiaries are not. Then, the Helper makes a choice between sharing their resources equally with the Beneficiary, or keeping all their resources to themselves. This choice is illustrated by the arrow labeled as "Share?". The Punisher observes the action of the Helper, and then makes a choice between punishing or not punishing the Helper when they do not share. This choice is illustrated by the arrow labeled as "Punish?" with the eye icon underneath the arrow emphasizing that the Punisher has observed the Helper's action before deciding. This punishment is costly to the Punisher; it requires them to forego one quarter of their resources in return for making the Helper who did not share lose half of theirs. The Helper is aware that the Punisher observes their

choice to share or not, and that the Punisher may choose to punish them if they do not share. Second, a new player—the Trustor—is paired with either the Helper or the Punisher, who observes either the Helper's or the Punisher's behavior in the third-party-punishment game depending on the experimental condition as illustrated by the arrow marked by an eye icon starting from the Trustor. All other players in the third-party-punishment game are aware of this fact (i.e., the existence of the Trustor and whom the Trustor is paired with). The Trustor then participates in a trust game where they are given resources that they may send to the Helper or to the Punisher depending on the experimental condition, knowing that any resources they send will be tripled, after which the Helper or the Punisher chooses how much of the tripled amount to return to the Trustor.

## People believe bots to behave as if they had preferences for money and other currencies

As a preliminary to our main series of studies, we assess the validity of our paradigm by explaining our stylized society (illustrated in Fig. 1) to 299 participants, checking that they correctly understood its rules and roles, and then asking them how much they agreed with a series of statements about the preferences of bots and humans. Questions and results are summarized in Fig. 2. Participants generally disagree that bots feel a desire or need for money in our paradigm, with an average rating of 25.7 (CI$_{95\%}$ = [22.212, 29.239]), significantly below the midpoint of the scale ($df$ = 298, $p$ < 0.001, t-statistic = −13.54, Cohen's $d$ = −0.78), and they disagree that bots feel a need or desire for likes on Twitter ($\mu$ = 32.8, CI$_{95\%}$ = [29.077, 36.428], $df$ = 298, $p$ < 0.001, t-statistic = −9.20, Cohen's $d$ = −0.53), or for not being banned on Wikipedia ($\mu$ = 29.9, CI$_{95\%}$ = [26.358, 33.502], $df$ = 298, $p$ < 0.001, t-statistic = −11.01, Cohen's $d$ = −0.64). In contrast, participants agree that humans feel a need or a desire for all these currencies. Money: $\mu$ = 83.1, CI$_{95\%}$ = [81.100, 85.141], $df$ = 298, $p$ < 0.001, t-statistic = 32.13, Cohen's $d$ = 1.89; Twitter likes: $\mu$ = 83.5, CI$_{95\%}$ = [81.185, 85.852], $df$ = 298, $p$ < 0.001, t-statistic = 28.15, Cohen's $d$ = 1.63; avoiding Wikipedia bans: $\mu$ = 81.9, CI$_{95\%}$ = [79.500, 84.312], $df$ = 298, $p$ < 0.001, t-statistic = 25.99, Cohen's $d$ = 1.50.

Most importantly though, participants agree that bots behave as if they had a preference for all these currencies, with agreement ratings significantly above the midpoint of the scale to which the following statistical tests refer. Money: $\mu$ = 64.6, CI$_{95\%}$ = [61.636, 67.502], $df$ = 298, $p$ < 0.001, t-statistic = 9.74, Cohen's $d$ = 0.56; Twitter likes: $\mu$ = 68.4, CI$_{95\%}$ = [65.126, 71.757], $df$ = 298, $p$ < 0.001, t-statistic = 10.90, Cohen's $d$ = 0.63; avoiding Wikipedia bans: $\mu$ = 64.0, CI$_{95\%}$ = [60.490, 67.484], $df$ = 298, $p$ < 0.001, t-statistic = 7.84, Cohen's $d$ = 0.45. As a result, we proceed with the assumption that the currency in our studies (money) is an adequate proxy for the currencies of real-life human-machine collectives online, such as likes and bans.

## Bots gain less trust than people by helping and punishing

Study 2 allows us to document the way people treat bots in our stylized society where agents can be Beneficiaries, Helpers, Punishers and Trustors (see Fig. 1 for an illustration of this society when all agents are humans, and see Methods for a detailed description of our experimental procedures). While only humans can be Trustors, the three other roles can be played by humans or bots in our experiments. This allows us to address three sets of research questions. The first set tracks changes in human behavior when bots are Beneficiaries, which we label with the letter **B** followed by a number for further reference:

**B1** Do people share with bot Beneficiaries as much as they share with humans?

**B2** When people share with bot Beneficiaries, do they receive the same trust-gain as when they share with humans?

**B3** When people do not share with bot Beneficiaries, are they punished to the same extent as when they do not share with humans?

**B4** When people punish human Helpers who do not share with bot Beneficiaries, do they receive the same trust-gain as when they punish human Helpers who do not share with human Beneficiaries?

The second set of research questions tracks changes in human behavior when bots are Helpers, which we label with an **H** followed by a number:

**H1** When bot Helpers share with human Beneficiaries, do they receive the same trust-gain as humans who share with human Beneficiaries?

**H2** When bot Helpers do not share with human Beneficiaries, are they punished to the same extent as humans in the same situation?

**H3** When people punish bot Helpers who do not share, do they receive the same trust-gain as when they punish human Helpers who do not share?

The last research question tracks changes in human behavior when bots are Punishers, which we label with a **P** followed by a number:

**P1** When bot Punishers punish human Helpers who do not share, do they receive the same trust-gain as humans who punish humans who do not share?

The results of Study 2 are presented in Fig. 3, and robustness analyses are described in the Methods section. We report the results of two-sided Welch's t-test to address unequal variances throughout, unless stated otherwise. The normality assumptions were formally tested, and not met, however, given the size of the sample we rely on t-tests as a convenient and practical approach[27]. We observe changes across almost all behaviors linked to bots being Beneficiaries. First (**B1**), people are significantly less likely to share with bot Beneficiaries (59%) than to share with humans (86%) ($df$ = 564.7, $p$ < 0.001, t-statistic = 7.94, Cohen's $d$ = 0.63, CI$_{95\%}$ = [0.204, 0.338], two-sided Welch's t-test). Second (**B2**), sharing with bots earns people a smaller trust-gain than sharing with humans: sharing with bots yields a 36 percentage points increase in trust, compared to a 54 percentage points increase when sharing with humans ($df$ = 614.3, $p$ < 0.001, t-statistic = 5.93, Cohen's $d$ = 0.48, CI$_{95\%}$ = [2.064, 24.013], two-sided Welch's t-test). This effect is largely driven by a greater leniency towards people who did not share with bots: these people inspire 27% trust, compared to only 14% for people who did not share with humans ($df$ = 553.9, $p$ < 0.001, t-statistic = −4.92, Cohen's $d$ = −0.40, CI$_{95\%}$ = [−17.942, −7.709], two-sided Welch's t-test). In contrast, we do not find credible evidence that people who shared with bots are trusted differently compared to people who shared with humans (63% vs. 68%, $df$ = 608.1, $p$ = 0.083, t-statistic = 1.74, Cohen's $d$ = 0.14, CI$_{95\%}$ = [−0.676, 11.102], two-sided Welch's t-test). Third (**B3**), people who did not share with bot Beneficiaries are less likely to be punished (27%) than people who did not share with humans (48%) ($df$ = 615.8, $p$ < 0.001, t-statistic = 5.608, Cohen's $d$ = 0.45, CI$_{95\%}$ = [0.138, 0.286], two-sided Welch's t-test). Fourth (**B4**), when people punish humans who did not share with bots, they earn a smaller trust-gain than when they punish humans who did not share with humans. Specifically, while the trust-gain is 21 percentage points when punishing those who do not share with humans, it is only 5 percentage points when punishing those who do not share with bots ($df$ = 628.0, $p$ < 0.001, t-statistic = 4.61, Cohen's $d$ = 0.37, CI$_{95\%}$ = [9.031, 22.442], two-sided Welch's t-test). In sum, we observe significant differences across almost all behaviors that involve bots as Beneficiaries, and it is tempting to see these as a behavioral cascade. One reason people do not share as much with bots (**B1**) might be that they get a smaller trust-gain for doing so (**B2**), and are less likely to be punished for not doing so (**B3**); and one reason they are less likely to be punished for not sharing is that this punishment earns the Punisher a smaller trust-gain (**B4**).

This cascade is born out in qualitative data, as all participants were asked to provide reasons for their decisions. There is no statistically significant difference in the share of participants who feared punishment when they were paired with bot Beneficiaries compared to humans, but fewer of them root their decisions in wanting to impress Trustors about their personal qualities or aim to strategically increase the amount of money they would be sent by Trustors (35% versus 30%, McNemar's test, $p$ < 0.001, $df$ = 1, OR = 1.982, CI$_{95\%}$ = [1.579, 2.534]), or reason that their decision was a result of whom they were as a person (e.g., trustworthy, 10% versus 4%, McNemar's test, $p$ < 0.001, $df$ = 1, OR = 9.677, CI$_{95\%}$ = [6.902, 15.459]). Importantly, in the human-only condition 47% of participants referenced higher-level principles (e.g., morality or ethics) for making their decisions, while in the condition

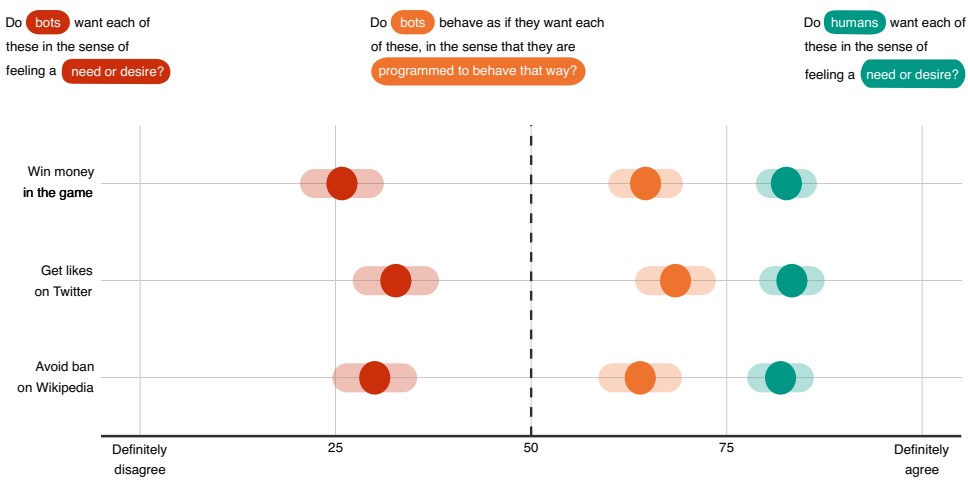

**Fig. 2 | Results of Study 1.** Data are presented as mean values and 95% confidence intervals. The number of participants in this study is 299.

**Fig. 3 | Results of Study 2.** The height of bars represent the mean values, the error bars represent standard deviations. In panel **a** the number of participants is 315 and 314 corresponding to the plotted order of bars from top to bottom. In panel **b** the number of participants is 314, 312 and 314, respectively. In panel **c** the number of participants is 318, 299 and 324, respectively. In panel **d** the number of participants is 312, 308, 318 and 313, respectively.

with bot Beneficiaries only 21% of them do so. For additional details see the Supplementary Information, Supplementary Note 8.

We now turn to our other research questions, examining behaviors linked to bots being Helpers or Punishers. First (**H1**), we observe that sharing with human Beneficiaries does not earn bots the same trust-gain as earned by human Helpers who shared with humans. Specifically, the trust-gain for bot Helpers is 45 percentage points, whereas the trust-gain for humans is 54 percentage points ($df = 638.4$, $p = 0.006$, t-statistic = 2.76, Cohen's $d = 0.22$, $CI_{95\%} = [2.504, 14.805]$, two-sided Welch's t-test). In contrast (**H2**), we find no credible evidence for punishment rates being different for bots who did not share with humans (42%) and humans who did not share with humans (48%) ($df = 625.9$, $p = 0.173$, t-statistic = 1.363, Cohen's $d = 0.11$, $CI_{95\%} = [-0.024, 0.132]$, two-sided Welch's t-test), nor (**H3**) for trust-gains being different for people who punished bots for not sharing (19 percentage points) and people who punished humans for not sharing (21 percentage points) ($df = 622.9$, $p = 0.639$, t-statistic = 0.469, Cohen's $d = 0.04$, $CI_{95\%} = [-5.097, 8.296]$, two-sided Welch's t-test).

Finally (**P1**), we find no credible evidence for different trust-gains between bots who punished human Helpers for not sharing (19 percentage points) and humans who punished other humans for not sharing (21 percentage points) ($df = 614.8$, $p = 0.678$, t-statistic = 0.415, Cohen's $d = 0.03$, $CI_{95\%} = [-5.048, 7.757]$, two-sided Welch's t-test). In sum, we observe little changes in the behaviors linked to bots being Helpers or Punishers, apart from the fact that bots do not earn the same trust-gain as humans when they share. The latter is also reflected in the qualitative responses of Trustors, who aimed to reward Helpers who shared or punish those who did not share when they were people (8%), but this justification implicating norms when Helpers were bots is less prevalent (4%, McNemar's test, $p < 0.001$, $df = 1$, OR = 11.923, $CI_{95\%} = [8.308, 20.131]$). Importantly, Trustors attribute more signaling value to the Helpers' behavior when they were people (41%), compared to when they were bots (31%, McNemar's test, $p < 0.001$, $df = 1$, OR = 1.702, $CI_{95\%} = [1.371, 2.141]$). For additional details see Supplementary Note 8.

In societies that only involve humans, people earn trust by sharing and by punishing those who do not share[23]. Such pro-social behaviors have been linked to the existence of consensual norms in prior work[28,29]. In the mixed human-bot collective that we investigate in Study 2, humans earn less trust when sharing with bots, bots gain less trust when sharing with humans, and humans gain less trust when punishing humans who did not share with bots. Studies 3, 4 and 5 seek to provide correlational and causal evidence that these lesser trust-gains follow, at least partially, from the uncertainty about consensual norms for sharing and punishing within mixed human-bot collectives.

**Trust gains depend on perceived helping and punishing norms**
Study 3 follows the general design of Study 2, with a few important differences. Specifically, participants only play the role of Trustors, and instead of making actual trust decisions they make them in a hypothetical scenario (without monetary compensation), but they answer three questions about norms (with financial incentives encouraging thoughtful guesses). The main objective of Study 3 is to show that the way people think about the appropriate behaviors in the third-party-punishment game (helping and punishing) relates to the trust-gain they confer to norm-followers over non-followers within experimental conditions. After familiarizing themselves with the two-stage game, Study 3 participants first estimate the proportion of Helpers who shared, or the proportion of Punishers who punished (depending on which experimental condition they were assigned to). Second, they state what they believe Helpers or Punishers should do (answering a yes/no question). Third, they estimate the proportion of participants who share these beliefs, and this third question about perceived normative consensus is our key predictor. Finally, participants state how much trust they would place in the Helper or the Punisher (again,

**Table 1 | The relationship between the Trustor's beliefs of norm-consensus over the Helper's sharing behavior and the trust that the Helper gains from sharing using multiple OLS regression**

| Variables | Estimate | p-value | Estimate | p-value |
|---|---|---|---|---|
| Consensus | 0.310 | $p < 0.001$ | 0.257 | $p < 0.001$ |
| | (0.228, 0.392) | | (0.133, 0.380) | |
| Norm | | | ✓ | |
| Empirical expectations | | | ✓ | |
| Fixed effects | | | ✓ | |
| Controls | | | ✓ | |
| Observations | 1075 | | 1075 | |
| Adjusted R-squared | 0.048 | | 0.065 | |

A ✓ indicates the inclusion of variables. Dependent variable: trust-gain of the Helper; Consensus: Trustor's guess of injunctive norm-consensus over the Helper's sharing; Norm: if one should share; Empirical expectations: the guessed proportion of Helpers who share; Fixed effects: fixed-effects for the experimental conditions; Controls: age, gender, race, education, income, and region. 95% confidence intervals are in parentheses. The complete regression table is Supplementary Table S5.

depending on the experimental condition). This design allows us to evaluate the relationship between the trust-gain of Helpers and Punishers when they follow norms (helping and punishing) as a function of Trustors' beliefs over the consensus of such norms.

Prior to turning to these results, however, we show descriptively that participants assume varying levels of consensus over (i) how Helpers and Punishers behave; and (ii) how Helpers and Punishers should behave (note related concepts such as second-order normative beliefs[30], and meta-norms[31]). Participants believe that there is more consensus over whether helping and punishing should be done, compared to the consensus they believe there is over actual helping and punishing acts, and without exceptions, a higher proportion of participants answer that Helpers should help and Punishers should punish in their respective conditions over their responses for what they believe the consensus behavior to be (see Supplementary Fig. S4). In other words, participants systematically underestimate the consensus across all conditions. The highest is the proportion of participants saying that the Helpers should share in their condition when no bots were involved. Interestingly, Trustors believe that people should punish bots for not sharing at the highest rate when expressing their views about the appropriateness of punishment, which should spark future research on peoples' perceptions over how bots should be programmed when interacting with humans.

The relationship between beliefs over norm-consensus and the trust-gain is presented in Tables 1 and 2, respectively, and they offer evidence that these beliefs lead Trustors to differentiate between their interaction partners who behave according to norms over not. The perceived consensus about the norm of sharing predicts the trust-gain of Helpers who shared over those who did not ($b_{OLS} = 0.257$, $CI_{95\%} = [0.133, 0.380]$, $p < 0.001$), robust to controlling for beliefs about actual behaviors, personal belief in the norm, fixed effects for experimental condition, and demographics. In other words, the stronger the consensus participants believe to be about sharing in their particular experimental condition determined by the identities of the Helper, Beneficiary and Punisher, the more they trust norm-followers over non-followers. Likewise, the perceived consensus about the norm of third-party punishment predicts the trust placed in Punishers who act against Helpers who did not share ($b_{OLS} = 0.109$ $CI_{95\%} = [0.004, 0.215]$, $p = 0.042$), again robust to controlling for beliefs about actual behaviors, personal belief in the norm, fixed effects for experimental condition, and demographics.

These results suggest that people attempt to navigate human-bot collectives by drawing on similar cooperative norms as in human

**Table 2 | The relationship between the Trustor's beliefs of norm-consensus over the Punisher's punishing behavior and the trust that the Punisher gains from punishing using multiple OLS regression**

| Variables | Estimate | p-value | Estimate | p-value |
|---|---|---|---|---|
| Consensus | 0.223 | $p < 0.001$ | 0.109 | $p = 0.042$ |
| | (0.152, 0.293) | | (0.004, 0.215) | |
| Norm | | | ✓ | |
| Empirical expectations | | | ✓ | |
| Fixed effects | | | ✓ | |
| Controls | | | ✓ | |
| Observations | 1439 | | 1439 | |
| Adjusted R-squared | 0.025 | | 0.065 | |

A ✓ indicates the inclusion of variables. Dependent variable: trust-gain of Punisher; Consensus: Trustor's guess of injunctive norm-consensus over Punisher's punishing; Norm: if one should punish for not sharing; Empirical expectations: the guessed proportion of Punishers who punish; Fixed effects: fixed-effects for the experimental conditions; Controls: age, gender, race, education, income, and region. 95% confidence intervals are in parentheses. The complete regression table is Supplementary Table S6.

collectives—only with the twist that they are more uncertain about how consensual the norms are about helping (and mostly about punishing) in this new context. Study 3 shows that people who believed in greater consensus are less likely to treat bots differently than humans. The logical next step is to attempt to inform people about how consensual the norms are, and to assess whether this information leads them to alter their behavior towards bots. This is what we accomplish in Studies 4 and 5. We choose to focus on the sharing (rather than the punishing) norm, for two practical reasons: Study 3 shows that the sharing norm is more consensual than the punishing norm in the human-only conditions, and the largest behavioral differences in Study 2 show a strong link to the trust earned by sharing. Taken together, these two observations make the sharing norm a more suitable candidate for observing a behavioral effect of norm-consensus information.

**Manipulating norm-consensus of helping may boost trust-gains**

Studies 4 and 5 follow the general design of Study 2, and serve the same purpose: both aim to causally link norm-consensus beliefs about the Helper's behavior on the trust they gain when they follow consensual norms over not following them. The studies have some key design differences, each with complementary strengths which we discuss in turn[32]. In Study 4 we invite the same Trustors who took part in Study 2, and assign them to the same experimental condition, with one important difference: before Trustors make their decision, they are informed that we recently conducted another similar study in which an overwhelming majority of participants (93%) said that Helpers should share with Beneficiaries in the specific condition they were assigned to (rather than in more general terms). This design makes norm-consensus salient at the point of the Trustors' decisions, and takes advantage of a within-person approach so that fixed characteristics of individuals which we have not measured are held constant across the two data collections by design. The effect of this information on the trust-gain of Helpers is displayed in Fig. 4a, where Trustors in Study 2 act as their own controls in Study 4. The consensus information affects the trust earned by bots who shared with humans: their trust-gain climbs to 55 percentage points, compared to 44 percentage points in Study 2 (df = 101, $p = 0.003$, t-statistic = 3.08, Cohen's $d = 0.31$, $CI_{95\%} = [4.256, 19.666]$, two-sided Welch's t-test).

Study 5 uses a between-person design, where Trustors are randomly assigned to either receiving a message about the norm-consensus, or receiving no message at all before they make their trust decisions. We inform them that we recently conducted another similar study in which the majority of participants said that Helpers should share with Beneficiaries, in the specific condition they were assigned to. This design makes norm-consensus salient at the time Trustors make their decisions using a between-person approach that alleviates concerns about learning from previous experiences. The effect of this information on the trust-gain of Helpers is displayed in Fig. 4b, where the two groups of Trustors recruited in Study 5 are compared to one another. The consensus information affects the trust earned by people who shared with bots: their trust-gain climbs to 46 percentage points, compared to 37 percentage points (df = 474.2, $p = 0.009$, t-statistic = 2.619, Cohen's $d = 0.23$, $CI_{95\%} = [2.261, 15.840]$, two-sided Welch's t-test).

Results from Studies 4 and 5 complement one another, see Supplementary Note 7, for an extensive meta-analysis. While the results from these two studies are different (one finds an effect in one condition, but not in the other and the findings in the second study are reversed), taken together, these studies show that our data are consistent with generally positive effects in case of bot Helpers (while the respective CI contains 0, leaving the possibility for a null-effect), and positive effects in case of human Helpers interacting with bot Beneficiaries. The results are also underscored by qualitative data where the norm-consensus information impacts the meaning assigned to Helpers decisions, in that Trustors are more likely to root their reasoning in the consistency of Helpers, and their qualities as actors (see Supplementary Note 8).

## Discussion

In five studies with 7917 individuals, and drawing on both quantitative and qualitative data, we explored how people navigate a mixed human-bot collective of Beneficiaries, Helpers, Punishers, and Trustors. Just as in human societies, we observed that actors earned trust by sharing and by punishing those who did not share—but we also observed that these trust-gains were less pronounced when bots are also part of their communities. For example, humans earned trust by sharing, but less so when they shared with bots; and humans earned trust by punishing, but less so when they punished bots. As a result, trust was not as easily established in our mixed human-bot collectives, for worse collective outcomes. Importantly though, trust-gains were only attenuated, rather than eliminated, suggesting that people carried into mixed human-bot collectives similar kind of assumptions about the social norms they relied on within human societies. We provided evidence that the perceived consensus about the norms of sharing and punishing was a driver of trust-gains. This was not only evident when analyzing trust decisions, but also substantiated when analyzing the reasons participants provided when explaining how they made their helping and trusting decisions. In addition, we followed up by demonstrating that trust-gains generally increased when informing participants of the high consensus about the norm of sharing, providing complementary evidence for this across two studies with different designs, suggesting that people might alter their behavior once informed.

It is generally accepted that stylized societies featuring incentivized interactions are an adequate way to capture the many other ways humans cooperate outside the lab. In a nutshell, humans want money, so money is a good currency for cooperation studies in the lab. Humans and bots cooperate in many ways outside the lab, too, but it is less clear that money is a good currency for human-bot cooperation studies in the lab, since bots have no use for money. Indeed, our participants said that bots had no desire for money, although they also said that bots acted as if they wanted money, or retweets. In other words, humans may have behaved in our stylized society in ways that are consistent with their expectations that bots would behave according to wanting money. The evidence presented here is consistent with people retweeting bots, or giving them money in stylized societies because prosociality toward bots is a good signal to send to other humans. Future work should evaluate if these results may be

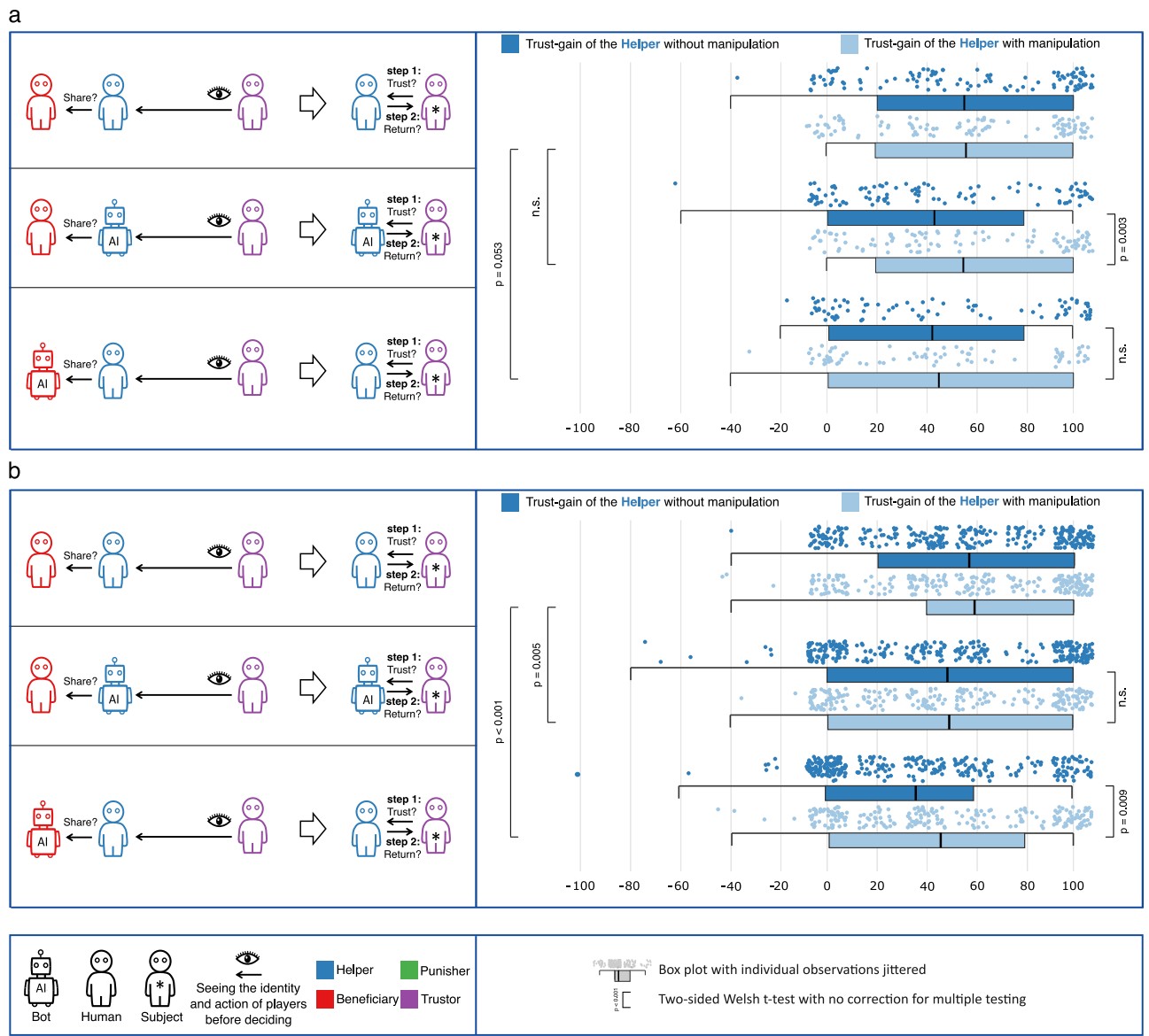

**Fig. 4 | Results of Study 4 and Study 5.** In panel **a** the number of participants is 111, 74, and 102 corresponding to the plotted order of pairs of box plots from top to bottom. In panel **b** the number of participants is 303, 251, 300, 223, 300 and 232 corresponding to the plotted order of box plots from top to bottom.

extended to various other situations of human-bot interactions to evaluate the scope conditions for these findings.

Our five studies come with the usual limitations of stylized experiments conducted with online samples. Our sample is large (with data from 7917 individuals) but not representative, since MTurk workers tend to be younger, more educated and more technologically savvy than the average American[33]. One possible upside, though, is that this sample may be closer to the population that is the most likely to currently interact with bots in online communities, with the downside being that these results may not generalize to older, less technologically savvy populations. Our reliance on a stylized society[23] allowed for tighter experimental control than if we had emulated the specific settings of one of these communities, but also means that our findings may play out differently depending on the idiosyncrasies of one platform or another. In this respect, it is important to note that the currency of cooperation will be different on each platform. Future research will have to carefully investigate how different currencies will impact the results we obtained using actual money. Social norms and expectations about humans and machines vary by context, culture

and lived experience[34,35], which limits the predictive value of a single set of studies. We provide evidence however using qualitative data showing that participants actively think about how their behavior toward bots is viewed and interpreted by other people. We also document that only one individual out of 3833 whose qualitative data we analyzed referenced the welfare of experimenters when thinking of the welfare of a bot, and that less than 4% of participants appeared confused or incredulous about the nature of their interaction partner. Therefore, explanations centered on confusion are unlikely to play a major role in our setting.

In sum, our findings provide a perspective on the future of human-machine collectives. We already knew that norms for prosocial behaviors were essential for cooperation in human societies[28,36-40]; our findings suggest that a similar mechanism can be leveraged for establishing trust within human-machine collectives. We show in particular that consensus about prosocial norms can emerge in these collectives, faster than people might expect—and that informing them about this emerging consensus could accelerate the pace at which trust is established within the collective.

## Methods

### Experimental procedures and measures

All five studies were programmed in Qualtrics, and follow the general structure described here: participants (1) read the consent form, and enter the study; (2) familiarize themselves with the rules of the game; (3) answer comprehension check questions; (4) make their choices followed by justifications for their choices; (5) answer manipulation check questions; and (6) answer some additional questions about their background. In all five studies, the information of whether a player is a bot or a human is communicated to participants via textual description referring to people as "MTurk worker" and to bots as "Bot," and also via a stylized image of either a robot or a person. This information is also presented on the screen where participants make their choices.

In Study 1 participants only answer questions about their beliefs of the desires and needs of a particular game participant in case they were a human or a bot, as well as additional questions about the same on Twitter and Wikipedia. Study 1 does not have any experimental conditions, and therefore random assignment. Participants in Studies 2, 3, and 5 are randomly assigned to experimental conditions using blinding, i.e., they are not aware of all the possible experimental conditions they could have been part of. In Study 3 participants make guesses which are incentivized using monetary incentives to encourage thoughtful decisions[41,42]. In Studies 2, 4 and 5 participants play behavioral games with monetary incentives, and receive bonuses on the basis of their decision and those of others in the game[24]. In Study 4 no randomization is used as participants are invited for the same condition they took part in in Study 2.

Helpers have a binary choice to share or not, i.e., they are not allowed to decide how much, and when sharing they split their resources evenly between themselves and Beneficiaries. Although sharing may seem to be motivated purely by altruism, Jordan and colleagues provide evidence that it may also serve a strategic purpose rooted in the incentives that are experienced in daily life and are made explicit in the game when interacting with Trustors[23] (see also earlier work on the reputational benefits, i.e., the signaling value of punishment[43,44]). Our interpretation of sharing is consistent with theirs (and while not our central focus, we demonstrate that these behaviors indeed signal trustworthiness, see Supplementary Fig. S5), and is confirmed by analyzing the reasons that participants give when making their helping decisions (for additional details, see Supplementary Note 8). We collect all Punishers' punishment decisions when Helpers do not share their resources. We collect all Trustors' sending decisions when Helpers share their resources and when they do not, and when Punishers punish a Helper who has not shared their resources and when they do not, depending on the experimental condition where Trustors are paired either with Helpers of Punishers. This use of the strategy method allows us to calculate five key quantities: (1) the percentage of all Helpers who share; (2) the percentage of all Punishers who punish Helpers who do not share; (3) the amount expressed in the % of their total resources that Trustors send for each possible choice of Helpers or each possible choice of Punishers depending on the experimental condition; and (4) the trust-gain of Helpers, which is the difference between the amount that Trustors send when a Helper shared and when they did not (i.e., differences in percentages); and (5) the trust-gain of Punishers which is the difference between the amount that Trustors send when a Punisher punished and when they did not (i.e., differences in percentages).

In Study 3 we follow exactly the design and procedures of Study 2, except that we now ask participants what they think other MTurk workers did and should have done, rather than have them make choices in the incentivized game. Specifically, we elicit (1) empirical expectations, i.e., the participants' beliefs about the expected choice of either Helpers or Punishers; (2) what they believe Helpers or Punishers should do; and finally (3) injunctive norms, i.e., how much consensus they believe there was among MTurk workers in the latter

matter[36,45]. Participants, therefore, are assigned to multiple experimental conditions: in three they respond to questions about Helpers, in four, they respond to questions about Punishers, which cover all possible configurations used in Study 2. Similarly to Study 2, the identity of Beneficiaries, Helpers and Punishers are signaled and reinforced with text and images. In sum, participants evaluate norms in their specific experimental conditions they are assigned to, rather than norms in more general terms about Helpers' and Punishers' behaviors. To encourage participants making thoughtful guesses, we offer a monetary reward to those who make correct guesses following recent experimental work on norm elicitation[41,42]. At the end of the experiment, participants answer a hypothetical question about how much trust they would place in the player they would have been paired with as a Trustor in the experimental condition they were assigned to. This allows us to connect beliefs about norm-consensus with the trust-gain.

In Study 4 we only recruit Trustors paired with Helpers and follow the same procedures outlined for Study 2, with the addition of a norm-consensus manipulation displayed on the screen where Trustors choose how much they trust the player they were paired with. Study 4 follows a within-person design where the decisions of participants without the norm-consensus signal are compared to decisions of the same people with the norm-consensus signal (made specific to the respective experimental condition), all else equal. The analytical sample contains participants who believe that the norm-consensus manipulation they received was truthful. The minimum number of days between participating in Study 2 and Study 4 was 356, while the average was 360.4, which partially addresses concerns about recall. To further alleviate these and other concerns common to within-person designs, we have included a detailed discussion and robustness analysis in Supplementary Note 2 and Supplementary Note 7.

In Study 5, to complement our quest in establishing a causal effect, we follow a between-person design. Participants are exposed to the same procedures as in Study 2, but they receive norm-consensus information stating that the majority of MTurk workers believed Helpers in their specific condition should help, rather than referring to a specific percentage (as in Study 4). The analytical sample contains participants who believe that the norm-consensus manipulation they received was truthful (78% over 63% in Study 2, which highlights one of the advantages of this design). Recruitment due to the turn-over of the population is not restricted to new participants, and in fact of the 2077 individuals who participated, only 1343 were people who have never taken part in any of the previous studies (64%). Those who have returned have a gap of a minimum 402 days between the two times they took part, with an average of 626.2 days, or almost two years. Robustness analyses also address repeat-participation, demonstrating substantively similar results when only non-returnees are analyzed (see Supplementary Fig. S10).

In all studies involving decision making we asked participants to justify their decisions after making them. From these responses we developed a set of qualitative codes concerning Helpers' and Trustors decisions, and four research assistants not familiar with the hypotheses coded 3833 responses. Each response was considered by two individuals, who then resolved their discrepancies. Supplementary Note 8 details our analyses of these data.

### Recruitment

We recruited participants on Amazon Mechanical Turk for all five studies using the services of CloudResearch (previously TurkPrime,[46]). For Study 1 we recruited about 300 participants (we started with 50 participants to ensure that the demographic filters worked correctly, and then followed up with 249 additional participants). For Study 2 the sample size was determined based on ref. [23]; consequently, about 300 participants in each role who showed adequate comprehension of the rules of the third-party punishment and trust games were recruited, yielding a total of 3761 individuals whose choices we analyze (note that

these also cover the human only conditions, i.e., all data presented in this paper have been part of our data collection). In Study 3 the sample size was determined based on Study 2, consequently, about 300 participants were recruited in the role of Trustors in each condition, yielding a total of 2514 individuals whose guesses we analyze. No participants in Study 2 were allowed to participate in Study 3; otherwise their experience in Study 2 may have influenced their guesses in Study 3. In Study 4 we invited the same participants who took part in Study 2 in the role of Trustor who were paired with Helpers. Therefore, all our analyses presented for Study 4 follow a within-person design; 458 individuals (48.67% of the sample of Trustors in Study 2) participated in Study 4. As detailed above, we analyze data from individuals who believed the norm-manipulation. Sample composition is described in Supplementary Table S7. In Study 5, we recruited on the same platform (to be able to track repeat participation, as many MTurk workers are also present on other similar platforms, like Prolific), but given the data quality restrictions applied, we did not eliminate prior participants from the pool of those who could take part, yielding an additional 1343 new individuals, and 734 returnees. The challenges posed by this approach are addressed with robustness analysis, see Supplementary Fig. S10.

**Inclusion criteria and data quality.** Only MTurk workers who were 18 years or older and who were located in the United States—as specified on their MTurk account and by their IP address—could see the "Human Intelligence Task" (HIT). To be eligible, workers also needed to have at least 100 HITs approved and a 95% approval rating. We also excluded workers from suspicious geolocations and those on the "universal exclude list," both managed by CloudResearch.

We used multiple screens to avoid the same individual participating in the study more than one time (i.e., between Study 2 and Study 3, as well as within all studies), making sure that all participants in a study saw only one of the experimental conditions. First, we created a survey group within CloudResearch for all HITs associated with this study. The same MTurk worker cannot take multiple surveys in the same CloudResearch survey group. Second, we activated the "ballot box stuffing" option in Qualtrics, to prevent multiple entries from the same IP address. In addition, we sequentially compiled a list of the MTurk IDs of workers who entered the study using an external SQL database and we automatically verified that each new worker was not already on this list. Our sample is limited to unique participants who took the study once (unless explicitly stated otherwise). Chronologically, Study 1 was conducted after Studies 2–5, and repeat entry in this study was possible to capitalize on data quality filters.

In Studies 1, 3, and 5 in addition to these filters, we showed the HIT only to MTurk workers who were "CloudResearch Approved Participants" to further enhance data quality using the services of CloudResearch. These filters alleviate concerns of data quality which were especially paramount at the time of data collection for Study 3[47,48], however they were not yet available at the time of conducting Study 2, which is why they were not employed. In addition to these filters, in Study 1, we collected data from Twitter users, which is possible via pre-screening filters managed by CloudResearch. Indeed, 98% of our sample are self-reported Twitter users as intended, and 93% of these participants use Twitter at least weekly.

Last, but not least, all five studies contain eight comprehension check questions: four related to the third-party punishment game, and four related to the trust game. In Study 1, participants were only allowed to proceed to the questions that form our main focus when they have answered all questions correctly. We allowed them multiple attempts. Study 2 and 5 participants who did not answer at least three of the four questions in each set correctly on two tries were not allowed to complete it. In Study 3 we did not screen out participants based on their comprehension, but they were only paid their bonus (i.e., their earning from the accurate guesses) if they showed the same level of comprehension as participants in Study 2, i.e., 75% of questions answered correctly, a measure which encourages careful attention. Study 4 participants are the same people who participated in Study 2, i.e., they have already shown adequate comprehension. In Study 4 the same rules applied to them as to the participants in Study 3, i.e., they only received their bonus from the game if they answered three out of the four comprehension check questions correctly in each set.

In addition to comprehension checks, we used manipulation checks in Studies 2–5 to make sure that participants were able to recall the treatment they participated in (i.e., participants were asked to identify the bot(s) if any in their experimental condition), and to gauge in Studies 4 and 5 if they believed the treatment, as people cannot be randomized to beliefs, and therefore we analyze the data of participants who were treated as intended. Our robustness analyses investigate if results on the full sample, and the restricted sample to participants with full comprehension and passing manipulation checks are consistent. The manipulation checks affected the sample in Studies 4 and 5 the most. In Study 4, interestingly, participants were less likely to accept the consensus information as true (52%) when presented with cases of humans sharing with bots, compared to the other cases. This might explain why the consensus information had a weaker effect on the trust earned by humans sharing with bots in this study—indeed, trust-gains were much larger (45 percentage points) for the 52% participants who believed in the consensus information than for the 48% participants who did not believe in the consensus (30 percentage points, $df = 138.9$, $p = 0.028$, t-statistic = 2.22, Cohen's $d = 0.37$, $CI_{95\%} = [1.627, 27.748]$, two-sided Welch's t-test). In Study 5, as our manipulation was more subtle (referring to the majority versus a specific share of people), 78% of participants believed that the information they received was truthful across all conditions, but variation still existed across conditions: participants were less likely to accept the consensus information as true when presented with cases of humans sharing with bots (74%), then, when presented with bots sharing with people (77%), and last when presented with the human only condition (84%).

## Ethical approval

All studies were approved by the NYU Abu Dhabi IRB (#062-2019), and informed consent was obtained from study participants consistent with the IRB protocol—Studies 1, 3, 4 and 5 were included as modifications to the IRB protocol filed for Study 2. In terms of deception, Study 2 and Study 3 did not use any, while Study 4 uses deception in two ways: (1) Helpers, Beneficiaries and Punishers were not invited for Study 4, and the earnings of Trustors were calculated by randomly matching Trustors to Helpers who participated in Study 2, i.e., Trustors' payoffs were calculated based on the decisions of real people, but their decisions did not impact the payoffs of real people; (2) the norm-consensus information we gave participants was only truthful in one of the three experimental conditions (the one that only involves people), which we obtained from Study 2. Everyone who participated in Study 4 was debriefed after the conclusion of data collection, consistent with study protocols. In Study 5, given the way we introduced to the norm-signal, no deception was necessary, and we recruited new individuals in the other roles whose payoffs were impacted by the average of Trustors' decisions.

## Sample composition and compensation

Data collection for Study 1 took place between 7–15 of June 2022, for Study 2 between 8–13 of August, 2019; for Study 3 between 14–19 of July, 2020; for Study 4 between 3–17 of August, 2020; and for Study 5 between 11–28 of September, 2021. Across the five studies we recruited 7917 individuals who completed one or more of the five studies. In Study 2, 2287 people failed to answer at least two comprehension check questions in either of the two sets on two attempts, whom we do not refer to as participants, and whose data we do not analyze (note

that they have never made decisions in the game). In this study we analyzed data from 3761 individuals who showed adequate comprehension. In Study 3, 2514 individuals participated, and 2066 people (82% of the sample) showed the same level of comprehension as those in Study 2. When restricting the sample in Study 3 to those with the same level of comprehension, we did not find any differences in demographic composition, see Supplementary Table S8. In Study 4, 458 individuals participated, 49% of the sub-sample of Trustors paired with Helpers in the trust game from Study 2. In Study 5 we recruited 2077 individuals, amongst whom 1343 were participants who have never seen any version of our studies. Of the remaining people, 378 took part in Study 2, 250 in Study 1 (which took place after Study 5), and 106 in Studies 2 and 4. The sample composition in Study 2 and Study 4 are slightly different, as the participants in Study 4 are slightly older, which likely represents patterns of turnover among MTurk Workers, see Supplementary Table S9.

Participants who completed Study 1 were 59% male (all other participants identified as female or other), and 69% identified as White (all other participants have identified as American Indian or Alaska Native, Asian or Asian American, Black or African American, Hispanic or Latino/a, Middle Eastern or North African, Other, or identified with multiple of these categories), with an average age of 38.2 (sd = 10.1). Participants who were allowed to complete Study 2 were 48% male, 75% identified as White, with an average age of 37.1 (sd = 11.8). Participants in Study 3 were 50% male, 70% identified as White, with an average age of 37.5 (sd = 12.6). Participants in Study 4 were 47% male, 76% identified as White, with an average age of 40.5 (sd = 12.4). Finally, participants in Study 5 were 47% male, 75% identified as White, with an average age of 41.3 (sd = 12.6). The average time to completion was 12.0 min in Study 1, 16.2 in Study 2, 15.8 min in Study 3, 15.2 min in Study 4, and 15.9 min in Study 5. The average earnings were $2.00, $1.79, $2.95, $4.97, and $1.47 in the five studies, respectively, yielding $10.00, $6.71, $11.20, $19.88, $5.55 in average hourly pay. Variation in these earnings is driven by multiple factors, one of which is the experimental condition. Whenever participants' payoffs were dependent on the bots' decisions, the bots were programmed to make these at random. Additional variation is driven by differences in show-up fees, which were set to $2.00 in Study 1 (note, in this study there were no incentives), $0.75 in Study 2, $2.00 in Study 3, $2.50 in Study 4, and $0.75 in Study 5. In addition to the show-up fee and bonus that participants could earn in Study 4, we selected five participants at random who were paid a $50.00 bonus on the MTurk platform. We chose to incentivize participants in addition to bonuses from the experimental game as we aimed to maximize the chances to recruit as many people as possible from Study 2 given the inferential strengths of a within-person design.

### Robustness checks
In Study 2 we tested the robustness of the results by excluding participants who (1) did not answer all comprehension check questions correctly (510 individuals, representing 14% of the sample); (2) failed the manipulation check, i.e., did not identify correctly the players who were bots in their experimental condition (575 individuals, representing 15% of the sample). We find that the results do not weaken despite the reduction in sample size, but strengthen in each case; see Supplementary Figs. S2 and S3.

In Study 3 we tested the robustness of the results by excluding participants who (1) did not answer all comprehension check questions correctly (1504 individuals, 60% of the sample); (2) failed to correctly identify who were bots in their experimental condition (617 individuals, 25% of the sample). In both robustness checks we find that the results are substantively similar, with some reduction in statistical significance due to the reductions in sample size; see Supplementary Note 1. We additionally compared the distribution of the trust-gain of Helpers and Punishers in all the experimental conditions in Study 2 to

Study 3, and found that they were substantively similar (see Supplementary Note 9) bolstering our confidence that the hypothetical decisions in Study 3 were (1) thoughtful, and (2) unlikely to have been the result of motivated reasoning[49].

In Study 4 we tested the robustness of the results by excluding participants who (1) failed to correctly identify who were bots in their experimental condition in either Study 2 or Study 4 (50 individuals, 17% of the sample); (2) added to the sample the participants who did not believe the experimental manipulation about norm-consensus over Helpers' behavior in their condition (171 individuals, 37% of the full sample), see Supplementary Note 2. All results remain similar, despite including individuals who did not believe the manipulation.

In Study 5 we also tested the robustness of the results by excluding participants who (1) failed to correctly identify who were bots in their experimental condition (238 individuals, 11% of the sample); (2) added to the sample the participants who did not believe the experimental manipulation about norm-consensus over Helpers' behavior in their condition (196 individuals, 9% of the full sample), see Supplementary Figs. S8 and S9. In case of the first robustness check, results remain similar, while in case of the second results are substantively similar, but no longer significant at the usual levels. In case of Study 5 we have also reproduced the results by dropping MTurk workers from the sample who have seen a version of this study previously, see Supplementary Fig. S10. These results are substantively similar.

We dedicate a supplementary note (Note 7) to meta-analyze Study 4 and Study 5 following Morris and DeShon[32], as their conclusions, analyzed separately are not identical (see Fig. 4). The upshot of this analysis is twofold: we find weak evidence for the homogeneity assumption, namely that the treatment effects are the same across the between- and within-person designs, which is not surprising, as these are two different theoretical quantities. When Helpers are bots, we find an increase in their trust-gain estimate with large variance, falling in the [−0.075, 0.387] 90% window, which is consistent with a large positive effect, but does not rule out a null-effect. When people help bot Beneficiaries, they benefit from the norm-signal as suggested by the [0.042, 0.248] 90% window of the trust-gain. Note however, that these combined results should be interpreted with caution due to lack of evidence that the homogeneity assumption holds.

Prior research identified selective attrition as a threat to causal identification when effect-heterogeneity is present[50]. Therefore, we calculated the percentage of participants who left the study after passing the comprehension check questions across experimental conditions, i.e., after they saw the treatment they were assigned to, yielding an average attrition rate across treatments of 2.74% in Study 2, 0.36% in Study 3, 0.23% in Study 4, and 0.67% in Study 5, respectively. We found no systematic differences across the conditions in their demographic composition within study (Supplementary Note 10), and therefore it is unlikely that selective attrition compromises data quality and our inferences.

### Pre-registration
The pre-registration of Study 2, took place on Open Science Framework (previously Experiments in Governance and Politics, egap.org) prior to the analysis of the outcome data (https://osf.io/cx5yv/, February 2, 2020). We pre-registered Study 3 and 4 prior to data collection (https://osf.io/3fhzn/, July 14, 2020 and https://osf.io/x53a4/, August 3, 2020, respectively). For Study 2 we registered an amendment after the analysis of the outcome data (https://osf.io/6r98y/, September 12, 2020).

Here, we discuss the amendment in detail (for all hypotheses, see Supplementary Fig. S1). The pre-analysis plan (PAP) originally contained 13 clearly stated hypotheses. In the amendment, we made two modifications to the PAP: on the one hand, we added four new hypotheses, all of which were direct consequences of hypotheses

already registered in the original PAP. On the other hand, we updated the directions for four hypotheses that were registered inconsistently with the rest. The omission of the four hypotheses was a flaw. They all keep to the following logic: we have registered hypotheses reflecting changes according to two processes. Specifically, Process 1: the level of trust when helping or punishing those who did not help is less compared to the baseline (H3a-1, H4a-1, H4b-1, and H4c-1); Process 2: the level of trust when not helping or not punishing those who did not help is more compared to the baseline (H3a-2, H4a-2, H4b-2, and H4c-2). These processes hold clear theoretical and testable implications for their difference, the trust-gain. H3a, H3b, H4a and H4b build on the already existing expectations in the original PAP (specifically H3a-1 and H3a-2; H3b-1 and H3b-2; H4a-1 and H4a-2; and H4b-1 and H4b-2, respectively) but were missing, and are contained in the amendment. These hypotheses are associated with the research questions **H1, B2, P1** and **B4** developed in the paper, respectively.

Second, during data analysis, we realized that some of the hypotheses were registered in an inconsistent direction to others. The seven experimental conditions in the PAP contain two baselines (depending on who participates in the trust-game: the Helper or the Punisher), and to these baselines we compare two and three additional conditions, respectively. Each of the manipulations in Study 2 entails manipulating the identity of players in the third-party punishment game proposed in the PAP. We anticipated in most of the conditions that these manipulations would decrease trust when sharing and third-party punishment were exhibited, and that trust would increase with the lack of sharing and lack of third-party punishment compared to the respective baselines. In four out of the 13 cases, the directions were registered inconsistently with these predictions, specifically: H3a-1, H3b-1, H3b-2, and H4c-2 which are associated with the research questions **H1, B2,** and **H3** developed in the paper, respectively.

### Reporting summary
Further information on research design is available in the Nature Portfolio Reporting Summary linked to this article.

## Data availability
All data analyzed in these studies have been deposited on Open Science Framework and can be accessed here[51]. At this repository pre-processed data can be found, and all analyses in this paper and its SI can be reproduced with these data. Pre-processing steps include (i) the removal of personally identifying information (i.e., MTurk IDs, and full qualitative responses, but not the codes, as they may contain identifying information together with other demographic data given by respondents); (ii) the merging of the data files collected in five different studies into a single file; (iii) the calculation of variables, such as, the trust gain (raw data that these calculations are based on are included in the data set).

## Code availability
No custom code were generated to analyze data collected for these studies. Standard techniques have been employed using the R statistical software (version: 3.6.3) and this code has been deposited together with the data here: https://osf.io/jzpvs/.

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

## Acknowledgements

We would like to thank Peter Bearman, Byungkyu Lee, and Mario Molina for their valuable comments, and the participants of the Indiana University Network Science Institute workshop for their insights and suggestions. We thank Katharina Klaunig, Irene Lin, Hannah Kasak-Gliboff, and Yao Xu for research support coding the qualitative responses. K.M. acknowledges the support of the Research Enhancement Funds received from NYUAD, funding from the NYUAD Center for Interacting Urban Networks (CITIES), funded by Tamkeen under the NYUAD Research Institute Award CG001; K.M. and T.R. acknowledge discretionary research support received from NYUAD; J.F.B acknowledges support from the Agence Nationale de la Recherche (ANR-19-PI3A-0004 and ANR-17-EURE-0010), and the research foundation TSE-Partnership.

## Author contributions

K.M., J.F.B and T.R. conceived the study and designed Study 1. K.M., and T.R. conceived the study and designed Study 2–5. W.L. also conceived and designed Study 2. K.M. created Study 1; K.M., W.L. and A.S. created Study 2; K.M. and A.S. created Study 3–5. K.M. performed data collection for Study 1. K.M., W.L. and A.S. performed data collection for Study 2; K.M. and A.S. performed data collection for Study 3–5. K.M., A.S., J.F.B. and T.R. produced the figures and tables; A.S. and K.M. created the Supplementary Information. K.M., T.R. and J.F.B. discussed the results and wrote the manuscript and revisions.

## Competing interests

The authors declare no competing interests.
