## [Peer Review File · Nature Communications]

Trust within human-machine collectives depends on the
perceived consensus about cooperative normsREVIEWER COMMENTS

Reviewer #1 (Remarks to the Author):

The paper reports the results of three studies examining how people interact with bots in a stylized decision-making environment. The first study finds that people share resources less with bots than with humans and that people get a smaller increase in trust (measured by the trust game) when they share resources with bots vs. humans. Two additional studies show that how much resources people share is correlated with perceptions of the social norm of sharing and that participants who participated in the study again but were told beforehand about the norm or sharing, shared resources more than they did the first time they participated in the study.

The topic examined in the paper is novel. The authors make a compelling case that people are increasingly interacting with bots on online platforms and that research is needed to understand these interactions.

Two main strengths of the paper are 1) the use of consequential decision-making, with monetary outcomes for the participants (although this was not spelled out in the manuscript, I assume this is the case, maybe make sure to mention this explicitly). 2) a comprehensive examination of bots in different roles, helpers, beneficiaries and so forth.

There are several points in the manuscript that can be strengthened and clarified.

First, although monetary decision-making tasks have a lot of advantages, it is unclear to me what participants think is the meaning of sharing resources with a bot. If I decide to give person A a few dollars, person A can use them to pay a bill, buy coffee, and so forth. The money has value for the person. It is unclear what type of value the money has for bots, and how people think about that (does the money go to the bot's creator?). In this sense, the stylized environment might be theoretically distinct from the online communities the authors describe in the introduction where the "coin" is content. From this perspective, it makes a lot of sense that people share resources less with bots, and are penalized less for not sharing with them, as giving money to an entity that cannot use it is not necessarily altruism or moral, it just might be an odd behavior. Importantly, this would be very different than a moderator censoring content created by a bot on social media or Wikipedia. I am not saying this is necessarily the case, and participants might think about this differently. However, it is plausible, and clarifying this issue could strengthen the paper.

Second, in Study 2, what are the differences between the perceived norms of sharing with humans vs bots? The study reports correlations between norms and behavior. However, according to the flow of the paper, I expected to see a comparison of sharing norms between bots and humans rather than a correlation across conditions.

Third, in Study 3 there is a confound. All participants went through the manipulation of learning about norms before participating again in the decision-making task. I share the author's intuition about the effectiveness of the manipulation, however, a stronger version of the study would use a control condition as well, such that half of the participants would learn about sharing-norms and half would not.

Two broader points:

I think the point about norms and consensus needs strengthening. We know norms affect behavior, that is not a novel. I think the interesting issue is why the norms are different. A difference in norms might be a by-product of people thinking about bots in a different way than they think about other humans, rather than the core difference. Some of the results in Study 3 are consistent with such an interpretation (e.g., people not believing that there's a norm of sharing with bots). The paper could be substantially strengthened, and more theoretically innovative if it addressed this issue.

The main text's brevity comes sometimes at the price of clarity. Information such as whether participants were paid according to their decision, how the bots were introduced and such are lacking. Since the word limit for the main text is 5,000 words, I would suggest the authors lengthen the

manuscript a bit more. If the authors decide to proceed with theorizing about norms, this could be better introduced in the introduction. Some of the statistical analyses could also be comprehensive. For example, conducting interaction analyses instead of mentioning that one comparison was significant and the other was not (page 4 lines 96-106). There are many more examples, but generally speaking, I think the manuscript would be easier to read and follow if the authors expanded a bit on their theorizing and studies.

Reviewer #2 (Remarks to the Author):

This paper seeks to test how people share with other humans compared with bots, and how others react to people to share with or punish humans compared with other bots. I like that the authors are examining the signaling aspects of helping & punishing, and how different acts bring different reputational benefits. The comparison of trust gains is a useful measure, and it's really interesting how people gain more reputation from helping humans than from helping bots – the reputational gains depend on who the beneficiary is (human vs. bot), but not so much on the actor is (human vs. bot).

There are some interesting questions here and a good methodology. However, there are several theoretical issues that need to be clarified, and the methods of Studies 2 & 3 need to be much clearer. Without more details on the methods, I cannot evaluate whether the results support the authors' main claims about how norms affect people's treatment of machines. The authors would need a strong revision to clarify this, with no guarantee of acceptance. So while it's possible that this paper could make a major contribution, there are several aspects that must be improved before that could be the case (especially points 1, 2, 5, and 6).

Theoretical/practical considerations:

1) I'd like to see more on the theoretical or practical justification for this work. Under what real-life conditions will humans be called upon to share money with bots? Or punish them? It seems a bit contrived to have people give money to a robot, or punish a robot by reducing its earnings. What does a bot care if it receives help or is punished – is it even programmed to notice? How does this impact its welfare, and why should anyone care? What real-life situations is this experiment meant to simulate, where people can help robots? There needs to be an explanation of the ecological relevance of this task, otherwise it comes across as a curiosity.

2) What motivation would a person have for sharing money with a robot or punishing it? When helping a person, a participant might help out of altruism (e.g., concern for the beneficiary), out of signaling (e.g., to reap trust gains), or out of confusion (e.g., see Max Burton-Chellew's many articles on confused cooperators). However, it's highly unlikely that people feel any altruism for the bots, or for any of the bot's payoffs for that matter. So it seems like the only real reason to help or punish a bot is for signaling purposes. (Though the authors should acknowledge Burton-Chellew's work about confusion – for example he shows that much conditional cooperation is simply people being confused about the game.) So how would the helping and punishment vary if it were all anonymous, such that signaling motives were lessened? I suspect there'd be much less helping and punishment of bots if it were anonymous, and the human-bot difference would be bigger.

3) There's a large literature on signaling and punishment that the authors have largely ignored, other than the Jordan et al 2016 article. For example, the earliest work by Barclay (2006, *Evol. Hum. Beh.*) shows that punishment is valued, but it's only valued if it's justified (i.e., against a valid target like a defector). If sharing with a bot isn't particularly valued, then it would be unjustified to punish someone for not sharing with a bot – those early results predict these current results well. Raihani & Bshary also have results that are relevant here. I encourage the authors to engage with previous work on the signaling value of punishment.

4) I believe there is also some previous work in experimental economics on people's treatment of computer players vs. human players. I don't remember this literature as well – I don't know if it's more than just an article or two – but it should be looked for and discussed. I'm sure that the previous work is not as comprehensive as the current study, so the current study would represent a major advance on that previous work. But I encourage the authors to dig up that previous work.

Methodological considerations:

5) I cannot evaluate Studies 2 and 3 because it's unclear what is happening. Is this the norm about any sharing, or sharing specifically with bots? We need more info on the methods to know what these results mean. For example, when given the manipulation about the consensus, is this the consensus about how humans should be treated, or how bots should be treated? If it's just the norms about any sharing (i.e., not specific to bots), then it seems largely irrelevant to how people treat bots differently from people. [On a theoretical note: there's no good reason why the norm of sharing with humans should also apply to sharing with bots – unlike humans, a bot does not care about what it receives, and in fact receives zero benefit from the sharing, so there's no good reason that a norm that applies to one should apply to the other.]

6) Because of point #4, it's not obvious how the authors can support their titular claim that trust "depends on the perceived consensus about cooperative norms". For example, it's not clear how the human-bot difference is affected by the perceived consensus – which specific result supports this conclusion? (Presumably it's supposed to be Tables 1 and 2, but they don't appear to say anything about humans vs. bots.) It's also not obvious which result supports this claim from the abstract: "we show that this behavior [differential treatment of bots vs humans] reflects their uncertainty about the consensual nature of cooperative norms when machines are involved". Also, some of the results contradict this claim in the abstract: "we show that informing them about this consensus decreases the differential treatment of machines." The results show that people who share with bots are not trusted any more, even after being given information on the consensus. In general, the authors need to be much clearer about what specific results support the claims about norms and the human-bot difference.

Minor points:

Table 1 and 2, are these humans vs bots or humans helping vs not helping?

In supplementary figure s5, should the participant be the helper and punisher instead of the trustor?

The terminology of (e.g.,) S-H(H)-B(H)-P(H) is unclear and doesn't always seem consistent. Perhaps this is because the same letters are used for multiple things: H can mean Human or Helper, and B can mean Beneficiary or Bot. I strongly recommend using different letters - or at least different cases - for one of these. It would help to give an example

It's unclear what Figure S7 and S8 are. These are the Standardized Mean Differences of some demographic variable? Which demographic variable? (Table S7 presents many.) What are the units?

Table S10: can this briefly state what a Bhattacharyya coefficient is? Many readers will not know what this is supposed to index. (A similar argument could be made for the eta there.) Let readers know easily what they're supposed to extract from this Table.

The word "Helper" is misspelled as "Heper" in the caption for Table S1, S2 and S5

Reviewer #3 (Remarks to the Author):

This paper tackles an important question about human-machine interaction and how to establish cooperation between them. Distrust in machines is well known, even when machines are objectively better and more able to help with decision-making (i.e. algorithm-aversion). However, the authors here ask to what extent similar mistrust may translate into cooperative domains and whether humans how trust them will get rewarded for their behavior.

This paper has several strong points. For instance, the authors use large sample sizes and good statistical analyses, and follow an established paper's procedures (in this case Jordan et al. 2016) which enables comparison and learning and essentially contribute to a collective replication effort of social science papers. The methods are solid and the results are very interesting.

My main concerns focus on the interpretation of the results. I'd suggest you think carefully about what participants were thinking of when interacting with bots (perhaps you have data to speak to these questions from open ended questions, or have additional experimental data) and how we should interpret their actions. I also have a few clarification questions around methods. Overall, I think this paper will make a nice contribution.

1) Why would machine have altruistic or strategic motives? What does it mean to share with a machine that can't own or possess anything?

You say (p. 10) that your "interpretation of sharing is consistent with" Jordan et al. but their study only involved humans and moral motives can be attributed to human decision-makers. It's much harder to argue the same for machines – why would you give to a machine who can't use the money for anything? And what motivation would a machine have to give to a human?

For example, I'm a bit puzzled why 60% of participants gave to the AI bot in Figure 2 – what did you tell them about the recipient that made them be so generous? How should we interpret this?

I like the idea of using economic games to study these questions with machines but it begs the question of how human participants *interpret* a machine's behavior or a machine's expectations of a human's behavior, and whether they think of anything that a machine has or does has involving social or strategic preferences

2) Point #1 is not merely philosophical but also important for the cognitive process underpinning the participants' decisions.

For example, do people think machines have "automated" social or strategic preferences (and if so, what are they) or do they think a programmer has instilled those preferences by defining how to act in certain kinds of situations - which means: what do people think those programmer's imputed social preferences into the machine are?

Your current set of studies primarily focus on norms and the uncertainty around them; my question here is to understand why there is uncertainty and around what.

3) Methods

A) I just wanted to better understand whether you used deception or not. Did machines actually share with some humans and not share with others? And what proportion of the machine sample did share versus not? Some more clarity would be helpful for future researchers.

B) Similarly, in Study 3, did you use deception or where did you get the 93% of previous participants

saying that Helpers should share with Beneficiaries? (Did the 52% or the 48% of people get it right that they did versus did not believe this information?)

C) I have more questions about Study 3 – maybe I missed this but why did you invite back participants from Study 1? Why not use a fresh sample? This seems to bring in a new host of questions about learning/remembering the previous study in these participants, and also follow-up questions about uptake rates and potentially selective uptake by observables or, more challenging, unobservables, which you'd have to explain and discuss.

Minor points:

1) I believe your robustness checks are very good and that your ITT analysis the right way to go. However, it is remarkable (in not a good way) that so many participants failed some sort of the comprehension check in all these studies (hundreds of participants, even 60% of the sample in Study 2). Luckily the results are not altered much whether they are in or out of the analysis but the take-away should be that these online platforms are getting increasingly problematic for research if these trends continue or get worse.

Put differently, despite your good practices, it is not encouraging that these comprehension failures exist and you may want to choose to run and replicate these results in a more controlled sample (e.g. lab) – however, I don't think it would be necessary for this paper as your results are consistent regardless of the inclusion criteria of your sample.

2) While the use of the term "consensus" is perfectly fine, you may be interested to know that other work in similar cooperative domains has referred to a similar concept as "second-order normative beliefs" (Jachimowicz et al. Nature Human Behaviour 2018) or "meta-norms" (Eriksson et al. Management and Organization Review 2017).

Dear Reviewers,

We hope you have stayed safe and healthy during these volatile times! We appreciate the invaluable opportunity to improve our work through the peer-review process. While we address the comments of Reviewers one-by-one, we here provide an overview of the most important changes we have made to the paper, addressing a few key observations that Editors and Reviewers converged on. Specifically:

1. As recommended, we conducted a between-participant version of Study 3. We did this in the new Study 4, recruited 2,077 participants, and derived substantively complementary findings to Study 3, which we detail in the revised manuscript (specifically in the updated Figure 3), strengthening our previous results. Beyond conducting a novel study, we also offer several robustness checks that address some of the shortcomings of Study 3, solidifying the implications of our previous findings. In addition to these, we conduct a meta-analysis of Study 3 and 4 which we detail in Supplementary Note 11, and reference in the main paper.
2. As requested, we elaborated on the fact that bot participants in our experiments have no use for money, which may create a methodological issue with the use of incentivized games. In a nutshell, we explain that bots have no use for anything, in the real world or in the lab, which means that using a currency they do not care for in the lab (money) does not create a disconnect from the real world, since bots have no use either for whatever the currency of cooperation is in real world applications.
3. We were asked to strengthen our narrative that the considerations of norms about sharing and punishment do feature prominently in Trustors' decision making. To achieve this, we carefully coded and analyzed the justifications that Trustors have given in Study 1, 3 and 4, in addition to what Helpers told us in Study 1. This required coding each justification (3810 responses, in total, ranging from a few words to a paragraph) by two independent research assistants each, who were not familiar with the paper and our hypotheses, and who then met and discussed each discrepant code, for the purposes of our analysis. Based on these new results, we added a section to the Supplementary Materials (Note 12), and made several changes to the main text. The results suggest that (1) Helpers are strategically thinking about Trustors' decisions; and (2) Trustors carefully consider Helpers' actions, especially when the third-party punishment game involves people, but also when it involves bots. We also demonstrate how these justifications shift with the norm-signals.

We hope the Reviewers agree that these changes are both substantial, as well as add to the body of evidence we have previously developed that perceptions about norms surrounding pro-social behaviors (such as sharing, and third-party punishment) in human-human, as well as human-bot collectives matter for the trust extended to norm-followers in contrast to non-followers, and that publicizing these norms contribute to a sharper distinction, leading to better collective outcomes overall. Next, we detail our responses to all Reviewer comments.

REVIEWER COMMENTS

Reviewer #1 (Remarks to the Author):

The paper reports the results of three studies examining how people interact with bots in a stylized decision-making environment. The first study finds that people share resources less with bots than with humans and that people get a smaller increase in trust (measured by the trust game) when they share resources with bots vs. humans. Two additional studies show that how much resources people share is correlated with perceptions of the social norm of sharing and that participants who participated in the study again but were told beforehand about the norm or sharing, shared resources more than they did the first time they participated in the study.

The topic examined in the paper is novel. The authors make a compelling case that people are increasingly interacting with bots on online platforms and that research is needed to understand these interactions.

Two main strengths of the paper are 1) the use of consequential decision-making, with monetary outcomes for the participants (although this was not spelled out in the manuscript, I assume this is the case, maybe make sure to mention this explicitly). 2) a comprehensive examination of bots in different roles, helpers, beneficiaries and so forth.

We thank the Reviewer for the suggestion. We now explicitly state in the “Experimental procedures and measures” subsection that monetary incentives were used. The added text specifically reads:

“In Study 1, 3 and 4 participants played behavioral games with monetary incentives, and received bonuses on the basis of their decision and those of others in the game [24]. In Study 2 participants made guesses which were also incentivized using monetary incentives to encourage thoughtful decisions [38, 39].”

There are several points in the manuscript that can be strengthened and clarified.

First, although monetary decision-making tasks have a lot of advantages, it is unclear to me what participants think is the meaning of sharing resources with a bot. If I decide to give person A a few dollars, person A can use them to pay a bill, buy coffee, and so forth. The money has value for the person. It is unclear what type of value the money has for bots, and how people think about that (does the money go to the bot’s creator?). In this sense, the stylized environment might be theoretically distinct from the online communities the authors describe in the introduction where the “coin” is content. From this perspective, it makes a lot of sense that people share resources less with bots, and are penalized less for not sharing with them, as giving money to an entity that cannot use it is not necessarily altruism or moral, it just might be an odd behavior. Importantly, this would be very different than a moderator censoring content created by a bot on social media or Wikipedia.

I am not saying this is necessarily the case, and participants might think about this differently. However, it is plausible, and clarifying this issue could strengthen the paper.

We agree with the Reviewer that bots do not care about money, and we also agree that what humans and bots exchange in online communities is not money. Community members can help others by upvoting or otherwise promoting their posts, by answering their questions, by defending them when they are attacked, just to mention a few. The key question is whether it makes sense to stylize this manifold of cooperation through a game with financial incentives. We first note that, in the offline world of humans, cooperation can be about much more than the exchange of money, too—but it is an accepted practice to use games with financial incentives as a way to capture this. We are building on this accepted, and widely-used practice when we use financial incentives as a way to stylize the manifold currencies of cooperation in human-bot online communities.

But is this stylized environment theoretically distinct from the online communities where the “coin” is content rather than money? What would be problematic is if bots did not care about the currency of our experiments (money) whereas they would actually care about the currency used in online communities (upvotes, bans, etc.). But this is not the case: bots do not care about anything, be it money or retweets. Everything is the same to them, which means that using money arguably does not create a problematic difference between our experiments and the settings they aim to capture.

Now the question might be: why would people help a bot, through sharing money or other means, if the bot does not care about it? Reviewer 2 nicely articulated the three reasons why they might do so: confusion about the game, altruism towards bots, or signaling to other humans. To elaborate:

- Confusion about the game is always possible despite our best efforts, and we do acknowledge it explicitly in the revised manuscript referencing recent literature;
- Altruism is unlikely since bots do not care about money—although some participants may consider that some humans (e.g., the ones who created or deployed the bot) indirectly benefit from the earnings of the bot;
- Signaling to other humans is presumably the best explanation of people’s generosity toward bots, and is indeed a main focus of our article, since one central measure of interest in our experiments is the trust gains, for example, that one can enjoy by helping bots or punishing those who do not help bots.

In sum, people can have various reasons to share with bots in our experiment, but the one we are especially interested in is signaling. If sharing money with bots is mostly a matter of making oneself look more trustworthy to other humans, then it does not matter how much bots care about money (if at all).

In the revised article, we provide a somewhat more compact version of the response above, because of word count limitations. Here's what we write:

“Unlike the experiment by Jordan and colleagues [23], the Beneficiaries, Helpers and Punishers in our experiments can be played by either bots or humans. Introducing bots in our stylized society raises a methodological and a theoretical question we both address upfront. From a methodological perspective, the issue at stake is that bots, unlike humans, do not care about money. This asymmetry would present a problem, if it did not exist in the real world that our stylized society tries to capture. That is, the fact that bots do not care about money would be a serious challenge to external validity, if the currency of human-bot cooperation in the real world (e.g., retweets, downvotes, blocking) was something that bots actually cared about. Bots, however, do not care about anything, be it money or retweets. As a result, the fact that bots are indifferent about the currency of cooperation in our stylized groups (money) is in line with them being indifferent about the currency of cooperation in the communities that we aim to model. The theoretical question, then, is why people may share resources with bots, money or otherwise. Altruism for such behavior is not a good explanation, since bots do not care about these resources (the reasoning which had been the justification of using bots in the earliest behavioral economics experiments studying motives other than altruism [25]). Confusion is always an option [26], although we will take extensive precautions so that human participants understand the incentive-structure of our stylized society. The explanation we focus on, instead, is signaling: humans help bots to signal to other humans that they are trustworthy. Evidence for this interpretation emerges from the analyses of both quantitative and qualitative data we collect and present in this paper.”

Second, in Study 2, what are the differences between the perceived norms of sharing with humans vs bots? The study reports correlations between norms and behavior. However, according to the flow of the paper, I expected to see a comparison of sharing norms between bots and humans rather than a correlation across conditions.

We thank the Reviewer for this comment, which highlighted a shortcoming of our presentation.

The motivation behind Study 2 has been to demonstrate that the trust-gain correlates with how clear Trustors believe the norms are about sharing and punishing. To put it differently, when they decide about trusting those who shared over those who did not share, or those who punished over those who did not, they base their decisions (consciously or otherwise) on how widely shared the norms they believed to be regulating the behavior of Helpers and Punishers. We thus added the following when we motivate Study 2:

“The main objective of Study 2 is to show that the way people think about the appropriate behaviors in the third-party punishment game (helping and punishing) relates to the trust-gain they confer to norm-followers over non-followers within experimental conditions. ”

We further agree with the Reviewer that the distribution of beliefs about norm-consensus is instructive, and therefore we now include such a distribution in the Supplementary Materials in SM Figure S6. Additionally, we write:

“Prior to turning to these results, however, we show descriptively that participants assumed varying levels of consensus over (i) how Helpers and Punishers behave; and (ii) how Helpers and Punishers should behave across experimental conditions. Participants believe that there is more consensus over whether helping and punishing should be done, compared to the consensus they believe there is over actual helping and punishing acts, and without exceptions, a higher proportion of participants answer that Helpers should help and Punishers should punish in their respective conditions over their responses for what they believe the consensus to be (see SM Figure S6). In other words, participants systematically underestimate the consensus across all conditions. The highest is the proportion of participants saying that the Helpers should have shared in their condition when no bots were involved.”

Third, in Study 3 there is a confound. All participants went through the manipulation of learning about norms before participating again in the decision-making task. I share the author’s intuition about the effectiveness of the manipulation, however, a stronger version of the study would use a control condition as well, such that half of the participants would learn about sharing-norms and half would not.

We appreciate the Reviewer’s point, and believe their concern is (partially) a result of the lack of clarity of our presentation, which we believe we have now strengthened in various places both in the main text and the Methods section. We briefly elaborate on this below.

Study 3 followed a within-person design where Study 1 responses of Trustors act as the control. The major advantage of this approach is that it removes the impact of all person-specific unobserved characteristics that do not change over time, but might impact trust-decisions. In this regard, a within-person design is highly desirable. An important shortcoming of this approach is that participants might remember their previous experience, which could influence their behavior at a later stage; for example, they could “anchor” on their previous decisions. If this were the case, our estimates would be conservative, as this would bias overall responses towards no change and would not work in our favor. We also believe that respondents were unlikely to remember their previous decisions or experience as the experiment took less than 15 minutes to complete and the average time between Study 1 and Study 3 was 250.6 days.

To reflect these points we write:

“Study 3 follows a within-person design where the decisions of participants without the norm-consensus signal are compared to decisions of the same people with the norm-consensus signal (made specific to the respective experimental condition), all else equal. The analytical sample analyzed contains participants who believed that the norm-consensus manipulation they received was truthful. The minimum number of days between participating in Study 1 and Study 3 was 212, while the average was 250.6, which partially addresses concerns about recall. To further alleviate these and other concerns common to within-person designs, we have included a detailed discussion and robustness analysis in SM Note 11.”

For the Reviewer’s convenience, we discuss the robustness analyses here as well.

“To address some of the shortcomings of the within-person design of Study 3, we performed the following robustness check. As we include a question in the experiment that asks about how the two studies (Studies 1 and 3) compare (specifically: “You have been invited to this study as a result of your participation in a study earlier. How do you think these two studies compare?”) with options presented on a 4-point scale ranging from the studies being identical to being completely different, and with a fifth option of “I don’t know as I do not remember the details of the previous study.” In order to alleviate concerns about recall, we carry out our analysis only on respondents who disclosed they did not remember the study (N = 172, 59.93% of the sample), and those who believed that the two studies were different (N = 21, 7.31% of the sample); in sum, we drop respondents who believed the studies were identical (which technically they were not, but they only differed in a single sentence, N = 10, 3.48% of the sample), and those who thought the studies were similar (N = 84, 29.27% of the sample). The results reported in Figure 3 in the main paper, now re-estimated on the smaller sample are presented in Figure S8 which offer the same substantive conclusions.”

Last but not least, the tension between a within-person and a between-person design emerges also in the comments of Reviewer 2. To fully alleviate these concerns we have now conducted a new study, Study 4, which addresses the same questions, but employing a between-person design. The details of this study can be found in (a) the main text, Results section; (b) the Methods section; (c) in the Supplementary Materials, Note 11. In a nutshell, we now present the results of both of these studies in Figure 3 with complementary findings accompanied by a meta analysis of these results.

Two broader points:

I think the point about norms and consensus needs strengthening. We know norms affect behavior, that is not a novel. I think the interesting issue is why the norms are different. A difference in norms might be a by-product of people thinking about bots in a different way than they think about other humans, rather than the core difference. Some of the results in Study 3 are consistent with such an interpretation (e.g., people not believing that there’s a norm of sharing with bots). The paper could be substantially strengthened, and more theoretically innovative if it addressed this issue.

We thank the Reviewer for this comment. We agree that what Helpers believe to be appropriate in terms of sharing does influence whether they share or not. We show, however, that what Trustors believe to be true about the norms governing *Helpers’ behaviors* influences their trust decisions, specifically, how much they differentiate between norm-followers and norm-breakers.

While norms are notoriously difficult to unpack, we have collected qualitative information from all respondents where we asked them to lay out the reasons for the decisions they have made. Specifically, we asked why Helpers shared or not, and why Trustors made their trust decisions. We coded these responses of Helpers’ and Trustors in both Study 1 (N = 1548), Study 3 (N = 458) and Study 4 (N = 1804), and summarized our procedures in the Supplementary Materials in

detail, but an abridged version of this is also included in the Methods section. The codes were developed by K.M., all responses were coded by two independent coders with discrepancies adjudicated between them who were not familiarized with the hypotheses being tested.

It is unlikely that respondents would formulate their reasoning in analytically precise terms used by sociologists or psychologists about norms. However, one piece of evidence for norms is when individuals fear punishment for not behaving according to them. For this reason, we specifically compared the share of respondents who reasoned through their Helping decisions in these terms, specifically, if they expressed that (a) they were concerned about Punishers punishing them and/or (b) if they expressed Trustors trusting them less or thinking lesser of them if they did not share. This helps us demonstrate that Helpers and Punishers conceptualized their decisions in normative terms across all conditions. Naturally, some justifications centered on other factors, such as bots not needing money which was referenced by 16.8% of Helpers. While these are interesting and important observations, our focus in this paper did not center on how consensus-beliefs emerge; but showing that they do influence behavior through motivations anchored on signaling, and that changing these beliefs would shift trust decisions accordingly.

Trustors mostly claimed they relied on the *consistency* of behaviors of Helpers across stages. Specifically 41.6% of Trustors claimed they expected more in return from Helpers who shared in the human only condition, in Study 1, which dwindles for both Helpers (31.3%), and people who help bots (27.8%). Few Trustors used the language of “reward” or “punishment” explicitly when describing their decisions (4.3%-8.3% in Study 1 across conditions). Yet, this language was applied more when Helpers were people instead of bots.

These points are now integrated into the manuscript, which to our knowledge is the first to discuss large-scale qualitative data from over 3800 people on justifications for human-bot interactions in a strategic behavioral setting.

The main text’s brevity comes sometimes at the price of clarity. Information such as whether participants were paid according to their decision, how the bots were introduced and such are lacking. Since the word limit for the main text is 5,000 words, I would suggest the authors lengthen the manuscript a bit more. If the authors decide to proceed with theorizing about norms, this could be better introduced in the introduction. Some of the statistical analyses could also be comprehensive. For example, conducting interaction analyses instead of mentioning that one comparison was significant and the other was not (page 4 lines 96-106). There are many more examples, but generally speaking, I think the manuscript would be easier to read and follow if the authors expanded a bit on their theorizing and studies.

We appreciate the suggestion of the Reviewer and have included various additional details throughout the manuscript. Most importantly, to also address some of the comments by the other two Reviewers we now motivate the studies more expansively, and have included more detailed discussion of some of the design choices, and measures.

Reviewer #2 (Remarks to the Author):

This paper seeks to test how people share with other humans compared with bots, and how others react to people to share with or punish humans compared with other bots. I like that the authors are examining the signaling aspects of helping & punishing, and how different acts bring different reputational benefits. The comparison of trust gains is a useful measure, and it's really interesting how people gain more reputation from helping humans than from helping bots – the reputational gains depend on who the beneficiary is (human vs. bot), but not so much on the actor is (human vs. bot).

There are some interesting questions here and a good methodology. However, there are several theoretical issues that need to be clarified, and the methods of Studies 2 & 3 need to be much clearer. Without more details on the methods, I cannot evaluate whether the results support the authors' main claims about how norms affect people's treatment of machines. The authors would need a strong revision to clarify this, with no guarantee of acceptance. So while it's possible that this paper could make a major contribution, there are several aspects that must be improved before that could be the case (especially points 1, 2, 5, and 6).

Theoretical/practical considerations:

1) I'd like to see more on the theoretical or practical justification for this work. Under what real-life conditions will humans be called upon to share money with bots? Or punish them? It seems a bit contrived to have people give money to a robot, or punish a robot by reducing its earnings. What does a bot care if it receives help or is punished – is it even programmed to notice? How does this impact its welfare, and why should anyone care? What real-life situations is this experiment meant to simulate, where people can help robots? There needs to be an explanation of the ecological relevance of this task, otherwise it comes across as a curiosity.

We thank the Reviewer for prompting us to elaborate on this issue. We agree that bots do not care about money, and we also agree that what humans and bots exchange in online communities is not money. In the introduction of the article, we provide ecological examples of humans and bots helping or hindering each other in online communities (references 4-14). Community members can help others by upvoting or otherwise promoting their posts, by answering their questions, by defending them when they are attacked, and more. The question is whether it makes sense to stylize this manifold of cooperation through a game with financial incentives. We first note that in the offline world of humans, cooperation can be about much more than the exchange of money, too—but it is an accepted practice to use games with financial incentives as a way to capture this manifold of cooperation. We are building on this accepted practice when we use financial incentives as a way to stylize the manifold currencies of cooperation in human-bot online communities.

We address this issue in the revised introduction, where we write:

“Unlike the experiment by Jordan and colleagues [23], the Beneficiaries, Helpers and Punishers in our experiments can be played by either bots or humans. Introducing bots in our stylized society raises a methodological and a theoretical question we both address upfront. From a

methodological perspective, the issue at stake is that bots, unlike humans, do not care about money. This asymmetry would present a problem, if it did not exist in the real world that our stylized society tries to capture. That is, the fact that bots do not care about money would be a serious challenge to external validity, if the currency of human-bot cooperation in the real world (e.g., retweets, downvotes, blocking) was something that bots actually cared about. Bots, however, do not care about anything, be it money or retweets. As a result, the fact that bots are indifferent about the currency of cooperation in our stylized groups (money) is in line with them being indifferent about the currency of cooperation in the communities that we aim to model. The theoretical question, then, is why people may share resources with bots, money or otherwise. Altruism for such behavior is not a good explanation, since bots do not care about these resources (the reasoning which had been the justification of using bots in the earliest behavioral economics experiments studying motives other than altruism [25]). Confusion is always an option [26], although we will take extensive precautions so that human participants understand the incentive-structure of our stylized society. The explanation we focus on, instead, is signaling: humans help bots to signal to other humans that they are trustworthy. Evidence for this interpretation emerges from the analyses of both quantitative and qualitative data we collect and present in this paper.”

2) What motivation would a person have for sharing money with a robot or punishing it? When helping a person, a participant might help out of altruism (e.g., concern for the beneficiary), out of signaling (e.g., to reap trust gains), or out of confusion (e.g., see Max Burton-Chellew’s many articles on confused cooperators). However, it’s highly unlikely that people feel any altruism for the bots, or for any of the bot’s payoffs for that matter. So it seems like the only real reason to help or punish a bot is for signaling purposes. (Though the authors should acknowledge Burton-Chellew’s work about confusion – for example he shows that much conditional cooperation is simply people being confused about the game.) So how would the helping and punishment vary if it were all anonymous, such that signaling motives were lessened? I suspect there’d be much less helping and punishment of bots if it were anonymous, and the human-bot difference would be bigger.

We thank the Reviewer for these comments. We are familiar with the work of Burton-Chellew and colleagues which demonstrates that even if respondents answer standard comprehension check questions correctly, they might not be able to determine the profit-maximizing strategy in a public goods game. Based on these observations, the authors conclude that a fair share of respondents are “confused.” We now have included a reference to this work, but also believe that the points made there have limited relevance in our case. Upon the coding of the qualitative responses, we demonstrate that many Helpers and Trustors articulate strategic motivations, and less than 4% of these participants reveal they have been confused about the rules of the two-stage game by in any condition indicating with their answer that they have not understood some aspect of the strategic interaction, or what information was available for decision makers. As we state in the Methods section we share the Reviewer’s assessment: *“Although sharing may seem to be motivated purely by altruism, Jordan and colleagues provide evidence that it may also serve a strategic purpose rooted in the incentives that are experienced in daily life and are made explicit in the game when interacting with Trustors [23]. Our interpretation of sharing is consistent with theirs, and is confirmed by analyzing the reasons that*

participants gave when making their helping decisions (for additional details, see SM Note 11). ”
To fully drive this point home, we now have included an analysis of the qualitative responses we collected showing why Helpers help and Trustors trust.

We also agree with the Reviewer’s speculation that if the signaling motivations were lessened (which could be achieved by the two stages being decoupled, and the first stage being examined in isolation), then Helping and Punishing behaviors would further decrease. However, since our focus in Studies 2-4 is on the trust-gain, such conditions would not help the main arch of our argument.

3) There’s a large literature on signaling and punishment that the authors have largely ignored, other than the Jordan et al 2016 article. For example, the earliest work by Barclay (2006, *Evol. Hum. Beh.*) shows that punishment is valued, but it’s only valued if it’s justified (i.e., against a valid target like a defector). If sharing with a bot isn’t particularly valued, then it would be unjustified to punish someone for not sharing with a bot – those early results predict these current results well. Raihani & Bshary also have results that are relevant here. I encourage the authors to engage with previous work on the signaling value of punishment.

We thank the Reviewer for pointing this out, and in fact, the papers the Reviewer suggests are all cited by Jordan and colleagues and form the basis of their experimental design. To make this link explicit, we included references to these papers in the Methods section where we discuss our interpretation of helping and punishment as a form of signaling.

4) I believe there is also some previous work in experimental economics on people’s treatment of computer players vs. human players. I don’t remember this literature as well – I don’t know if it’s more than just an article or two – but it should be looked for and discussed. I’m sure that the previous work is not as comprehensive as the current study, so the current study would represent a major advance on that previous work. But I encourage the authors to dig up that previous work.

We thank the Reviewer for this suggestion. We have now included citations pointing to this work. Specifically, we mention that including computerized players has been done to eliminate altruism as a motivation, with the aim to measure confusion. However, we believe that our study highlights that other motivations might also drive behavior, such as assumed norms, and therefore, reputational concerns. Our two-stage game (building on the work of Jordan and colleagues) makes these explicit.

Methodological considerations:

5) I cannot evaluate Studies 2 and 3 because it’s unclear what is happening. Is this the norm about any sharing, or sharing specifically with bots? We need more info on the methods to know what these results mean.

We thank the Reviewer for these suggestions, which we believe have made our writing clearer.

To address these concerns (also echoed by Reviewer 1) we provided more details. Specifically, we now write in the Methods section for Study 2 the following: *“Participants, therefore, are assigned to multiple experimental conditions: in three they responded to questions about Helpers, in four, they responded to questions about Punishers, which cover all possible configurations used in Study 1. Similarly to Study 1, the identity of Beneficiaries, Helpers and Punishers is signaled and reinforced with text and images. In sum, participants evaluate norms in their specific experimental conditions they are assigned to, rather than norms in more general terms about Helpers’ and Punishers’ behaviors.”* We have also revised the text in the Results section significantly to provide more clarity and emphasize that all respondents evaluated measures of norms specific to their experimental context signaled by the identity of Beneficiaries, Helpers and Punishers.

We also made several changes to the Results section discussing Study 3, and clarified that the norm-consensus information that respondents received was also made specific to the experimental condition they were assigned to. In the Methods section we now added:

“Study 3 follows a within-person design where the decisions of participants without the norm-consensus signal are compared to decisions of the same people with the norm-consensus signal (made specific to the respective experimental condition), all else equal. The analytical sample analyzed contains participants who believed that the norm-consensus manipulation they received was truthful. The minimum number of days between participating in Study 1 and Study 3 was 212, while the average was 250.6, which partially addresses concerns about recall. To further alleviate these and other concerns common to within-person designs, we have included a detailed discussion and robustness analysis in SM Note 11.”

We hope this addition addresses the concerns of the Reviewer (also echoed by Reviewer 1) where we provided even further details and discussion of the strengths and weaknesses of a within-person approach, and explained the details of Study 4 to further bolster our findings.

For example, when given the manipulation about the consensus, is this the consensus about how humans should be treated, or how bots should be treated? If it’s just the norms about any sharing (i.e., not specific to bots), then it seems largely irrelevant to how people treat bots differently from people. [On a theoretical note: there’s no good reason why the norm of sharing with humans should also apply to sharing with bots – unlike humans, a bot does not care about what it receives, and in fact receives zero benefit from the sharing, so there’s no good reason that a norm that applies to one should apply to the other.]

We thank the Reviewer for pointing this out, as it gave us an opportunity to improve our writing. We agree that a general norm about any sharing would not be the right approach, and this is not the approach we took. In every instance, the norm is specific to the actions of the Helper or Punisher *in the specific experimental condition* signaling the identities of all players. We also agree with the Reviewer’s remark that there is no good theoretical reason why a sharing-related norm should or would be extended to bots in isolation, i.e., in the third-party punishment game alone (of course save from impression management of the experimenter). Our goal with the experimental design is to show that *when* acts of sharing do not take place in

isolation, *then* norms assumed to surround such acts influence levels of trust placed in those who abide by the norms over those who do not.

6) Because of point #4, it's not obvious how the authors can support their titular claim that trust "depends on the perceived consensus about cooperative norms". For example, it's not clear how the human-bot difference is affected by the perceived consensus – which specific result supports this conclusion? (Presumably it's supposed to be Tables 1 and 2, but they don't appear to say anything about humans vs. bots.) It's also not obvious which result supports this claim from the abstract: "we show that this behavior [differential treatment of bots vs humans] reflects their uncertainty about the consensual nature of cooperative norms when machines are involved". Also, some of the results contradict this claim in the abstract: "we show that informing them about this consensus decreases the differential treatment of machines." The results show that people who share with bots are not trusted any more, even after being given information on the consensus.

In general, the authors need to be much clearer about what specific results support the claims about norms and the human-bot difference.

We especially thank the Reviewer for this point, and we agree that the original manuscript lacked clarity in this regard. We have substantially revised the manuscript throughout to improve on this point.

In the abstract we now write: *"We also demonstrate that the trust gain of norm-followers is associated with trustors' assessment about the consensual nature of cooperative norms over helping and punishing. Lastly, we establish that informing trustors about the norm-consensus over helping generally decreases the differential treatment of machines, and people interacting with them."*

When discussing Tables 1-2 we also improved the text, specifically adding the following details:

"The main objective of Study 2 is to show that the way people think about the appropriate behaviors in the third-party punishment game (helping and punishing) relate to the trust-gain they confer to norm-followers over non-followers within experimental conditions."

We also write:

"This design allows us to evaluate the relationship between the trust-gain of Helpers and Punishers when they follow norms (helping and punishing) as a function of Trustors' beliefs over the consensus of such norms. Prior to turning to these results, however, we show descriptively that participants assumed varying levels of consensus over (i) how Helpers and Punishers behave; and (ii) how Helpers and Punishers should behave across experimental conditions. Participants believe that there is more consensus over whether helping and punishing should be done, compared to the consensus they believe there is over actual helping and punishing acts, and without exceptions, a higher proportion of participants answer that Helpers should help and Punishers should punish in their respective conditions over their responses for what they believe

the consensus to be (see SM Figure S6). In other words, participants systematically underestimate the consensus across all conditions. The highest is the proportion of participants saying that the Helpers should have shared in their condition when no bots were involved. Interestingly, Trusters believe that people should punish bots for not sharing at the highest rate when expressing their views about the appropriateness of punishment, which should spark future research on peoples' perceptions over how bots should be programmed when interacting with humans."

Study 3 and Study 4 (the latter of which we conducted specifically for this review) are the ones that speak specifically to the point that norm-consensus signals shift behaviors. We believe our writing now reflects these key distinctions throughout, from the Abstract to the Discussion.

Minor points:

Table 1 and 2, are these humans vs bots or humans helping vs not helping?

Both tables contain all conditions, and specifically fixed effects models as the table captions describe. As a consequence, these results establish that beliefs about norm-consensus correlate with trust-gain consistently within experimental conditions.

In supplementary figure s5, should the participant be the helper and punisher instead of the trustor?

We agree with the Reviewer, and believe that the title of the figure suggests so: "Sharing and third-party punishment is a costly signal of trustworthiness, regardless of the identity of players." Note, this is currently Figure S10 after the revision.

The terminology of (e.g.,) S-H(H)-B(H)-P(H) is unclear and doesn't always seem consistent. Perhaps this is because the same letters are used for multiple things: H can mean Human or Helper, and B can mean Beneficiary or Bot. I strongly recommend using different letters - or at least different cases - for one of these. It would help to give an example it's unclear what Figure S7 and S8 are. These are the Standardized Mean Differences of some demographic variable? Which demographic variable? (Table S7 presents many.) What are the units?

We agree with the Reviewer and follow the naming convention we used in the experiment that used the natural terms of Player 1 for Helper, Player 2 for Beneficiary, Player 3 for Punisher, and Player 4 for Trusters. We further added an explanatory paragraph to both Supplementary Notes, list the variables used to calculate the SMDs, and specify the units in the figures.

Table S10: can this briefly state what a Bhattacharyya coefficient is? Many readers will not know what this is supposed to index. (A similar argument could be made for the eta there.) Let readers know easily what they're supposed to extract from this Table.

We thank the Reviewer for pointing out this shortcoming, and we have now included these details in Supplementary Note 13.

The word “Helper” is misspelled as “Heper” in the caption for Table S1, S2 and S5

Thanks for spotting this typo; we have now corrected it.

Reviewer #3 (Remarks to the Author):

This paper tackles an important question about human-machine interaction and how to establish cooperation between them. Distrust in machines is well known, even when machines are objectively better and more able to help with decision-making (i.e. algorithm-aversion). However, the authors here ask to what extent similar mistrust may translate into cooperative domains and whether humans how trust them will get rewarded for their behavior.

This paper has several strong points. For instance, the authors use large sample sizes and good statistical analyses, and follow an established paper’s procedures (in this case Jordan et al. 2016) which enables comparison and learning and essentially contribute to a collective replication effort of social science papers. The methods are solid and the results are very interesting.

My main concerns focus on the interpretation of the results. I’d suggest you think carefully about what participants were thinking of when interacting with bots (perhaps you have data to speak to these questions from open ended questions, or have additional experimental data) and how we should interpret their actions. I also have a few clarification questions around methods. Overall, I think this paper will make a nice contribution.

We appreciate the suggestion of the Reviewer to bring additional data to bear on the question of what participants were thinking when making their decisions. Fortunately, we indeed collected such data through all studies involving strategic decision making discussed in the paper, as well as the novel study we carried out for the purposes of this review. We have now included an analysis of respondents’ qualitative answers justifying their actions, which we hope the Reviewer will agree makes for a much stronger contribution, as ours is the first to report on such data in a large-scale behavioral experiment.

As we suspected, participants have mostly been thinking about other people when interacting with bots. We discuss these data which we collected in Study 1, Study 3 and Study 4 at length in the Results section, the Methods section, and Supplementary Materials (Note 11). We hope these additional analyses bolster the interpretation of the results.

1) Why would machines have altruistic or strategic motives? What does it mean to share with a machine that can’t own or possess anything?

We agree that it makes little sense to say that a machine would have altruistic or strategic motives: machines do not have such motivations (or any other motivations). But machines can

display strategic or altruistic behavior, as long as they are programmed to do so, and we do not mean anything more than that in our manuscript.

We also agree that it is unclear what it means to share money with a machine that cannot own anything. Reviewer 2 nicely articulated the three reasons why people might do so (which we incorporated into the paper and address):

- Altruism is unlikely since bots do not care about money—although some participants may consider that some humans (e.g., the ones who created or deployed the bot) indirectly benefit from the earnings of the bot, a common explanation in behavioral economics, which we have only very limited evidence for (among the 3810 justifications we analyzed, only 2 mentioned the experimenter).
- Confusion about the game is always possible despite our best efforts, and we do acknowledge it in the revised manuscript (and add that we found less than 4% of justifications which pointed to the respondent being confused);
- Signaling to other humans is presumably the best explanation of people's generosity toward bots, and is indeed a main focus of our article, since one central measure of interest in our experiments is the trust gains that one can enjoy by helping bots or punishing those who do not help bots.

In sum, people can have various reasons to share with bots, but the one we are especially interested in is signaling, i.e., sharing money with bots as a way of making oneself look more trustworthy to other humans.

We discuss these issues in the revised introduction of the article, where we write :

“Unlike the experiment by Jordan and colleagues [23], the Beneficiaries, Helpers and Punishers in our experiments can be played by either bots or humans. Introducing bots in our stylized society raises a methodological and a theoretical question we both address upfront. From a methodological perspective, the issue at stake is that bots, unlike humans, do not care about money. This asymmetry would present a problem, if it did not exist in the real world that our stylized society tries to capture. That is, the fact that bots do not care about money would be a serious challenge to external validity, if the currency of human-bot cooperation in the real world (e.g., retweets, downvotes, blocking) was something that bots actually cared about. Bots, however, do not care about anything, be it money or retweets. As a result, the fact that bots are indifferent about the currency of cooperation in our stylized groups (money) is in line with them being indifferent about the currency of cooperation in the communities that we aim to model. The theoretical question, then, is why people may share resources with bots, money or otherwise. Altruism for such behavior is not a good explanation, since bots do not care about these resources (the reasoning which had been the justification of using bots in the earliest behavioral economics experiments studying motives other than altruism [25]). Confusion is always an option [26], although we will take extensive precautions so that human participants understand the incentive-structure of our stylized society. The explanation we focus on, instead, is signaling: humans help bots to signal to other humans that they are trustworthy. Evidence for

this interpretation emerges from the analyses of both quantitative and qualitative data we collect and present in this paper."

You say (p. 10) that your "interpretation of sharing is consistent with" Jordan et al. but their study only involved humans and moral motives can be attributed to human decision-makers. It's much harder to argue the same for machines – why would you give to a machine who can't use the money for anything? And what motivation would a machine have to give to a human?

We thank the Reviewer for this question as it allows us to articulate in our response, and more clearly in the manuscript, that our focus is on the decisions *made by humans in human-bot collectives*, as opposed to how bots should be programmed to act in them. These decisions are influenced by strategic considerations, specifically, how their actions are viewed (among other factors).

For example, I'm a bit puzzled why 60% of participants gave to the AI bot in Figure 2 – what did you tell them about the recipient that made them be so generous? How should we interpret this?

We thank the Reviewer for this question, as it highlights that our writing has not made it clear enough that Helpers or Punishers took part in a two-stage interaction, i.e., the behavior in Stage 1 should not be considered in isolation. As Reviewer 1 pointed out, a third-party punishment game played in isolation would have yielded different levels of helping and punishment behaviors, and the contrast between experimental conditions would have likely been even starker. However, given that people who participated in Stage 1 were anticipating Stage 2 (just as in the case of the work by Jordan and colleagues) they made their decisions in this strategic context.

I like the idea of using economic games to study these questions with machines but it begs the question of how human participants **interpret** a machine's behavior or a machine's expectations of a human's behavior, and whether they think of anything that a machine has or does has involving social or strategic preferences.

This comment of the Reviewer echoes comments made by other Reviewers. To address this, we now include detailed analyses of participants' justifications for the decisions they have made to clarify this. The details are included in the Results section and in Supplementary Materials Note 11.

2) Point #1 is not merely philosophical but also important for the cognitive process underpinning the participants' decisions.

For example, do people think machines have "automated" social or strategic preferences (and if so, what are they) or do they think a programmer has instilled those preferences by defining how to act in certain kinds of situations - which means: what do people think those programmer's imputed social preferences into the machine are?

In the qualitative responses, some respondents (about 12%, averaging over Trustors' justifications) stated that they did not know how machines make decisions. Some thought machines make *random* decisions (about 2%), and therefore they reason, no inferences can be made on the basis of what they might do from one situation to another. Others believed that machines are programmed to act consistently across situations. These perspectives highlight, again, that respondents were not considering the stages of the game in isolation. It is a fascinating and open question for future research: what people assume across a broad range of different contexts about the principles of how bots make decisions, which points beyond the scope of our study. However, our analyses provide a starting point for such research grounded in the justifications of people who interact with bots.

Your current set of studies primarily focus on norms and the uncertainty around them; my question here is to understand why there is uncertainty and around what.

This question is deep and general: why do we have uncertainty about norms in some settings compared to others? Likely, this relates to how the behavior and its enforcement are common and visible to others, which fall outside of our scope. However, we also believe that the analysis of the qualitative data provides us with some answers. Trustors appear to be focusing on the *meaning* of Helpers' actions, and how clearly they might be able to interpret them across the different scenarios. This bolsters the speculation above, but future work is needed to make a conclusive assertion.

3) Methods

A) I just wanted to better understand whether you used deception or not. Did machines actually share with some humans and not share with others? And what proportion of the machine sample did share versus not? Some more clarity would be helpful for future researchers.

In the Method section we state: *"Whenever participants' payoffs were dependent on the bots' decisions, the bots were programmed to make these decisions at random."* We have not made any statements about how bots *would* make their decision, and we have not programmed decisions on the basis of humans' actions in the respective experimental conditions.

B) Similarly, in Study 3, did you use deception or where did you get the 93% of previous participants saying that Helpers should share with Beneficiaries? (Did the 52% or the 48% of people get it right that they did versus did not believe this information?)

These details were stated in the section about ethical approval. Specifically: *"Study 1 and Study 2 did not use any form of deception. Study 3 uses deception in two ways: (1) Helpers, Beneficiaries and Punishers were not invited for Study 3, and the earnings of Trustors were calculated by randomly matching Trustors to Helpers who participated in Study 1, i.e., Trustors' payoffs were calculated based on the decisions of real people, but their decisions did not impact the payoffs of real people in Study 3; (2) the norm-consensus information we gave participants was only truthful in one of the three experimental conditions (the one that only involves people),*

which we obtained from Study 2. Everyone who participated in Study 3 was debriefed after the conclusion of data collection, consistent with study protocols."

To avoid the use of deception (as well as to address other concerns) in Study 4 we used a slightly different manipulation, specifically we stated that "the majority" of previous participants stated that Helpers should help in the respective scenarios. This new information is added now to the section referenced above.

C) I have more questions about Study 3—maybe I missed this but why did you invite back participants from Study 1? Why not use a fresh sample? This seems to bring in a new host of questions about learning/remembering the previous study in these participants, and also follow-up questions about uptake rates and potentially selective uptake by observables or, more challenging, unobservables, which you'd have to explain and discuss.

We appreciate these points of the Reviewer, and agree that a within-person design has shortcomings. We now include a longer discussion of our design choices, alongside a novel robustness analysis which we have detailed in response to Reviewer 1's similar concerns. An explicit discussion of all these issues is now included in Supplementary Note 11.

In addition, however, we also chose to repeat Study 3 using a between-person design by recruiting a new large sample (N = 2077 individuals) which we discuss in the Methods and Results section. We believe that the evidence from Study 3 and the evidence from Study 4 are complementary. Taken together, they bolster the claim that manipulating norm-consensus information enhances the trust-gain of norm-followers, even when the norm-followers are bots or when the norm-followers' actions impact bots. However, we also document that successful norm-manipulation may not always be achieved. To reflect these nuances we made sure to clarify the language describing these findings in the Abstract as well.

Minor points:

1) I believe your robustness checks are very good and that your ITT analysis the right way to go. However, it is remarkable (in not a good way) that so many participants failed some sort of the comprehension check in all these studies (hundreds of participants, even 60% of the sample in Study 2). Luckily the results are not altered much whether they are in or out of the analysis but the take-away should be that these online platforms are getting increasingly problematic for research if these trends continue or get worse.

Put differently, despite your good practices, it is not encouraging that these comprehension failures exist and you may want to choose to run and replicate these results in a more controlled sample (e.g. lab) – however, I don't think it would be necessary for this paper as your results are consistent regardless of the inclusion criteria of your sample.

We thank the Reviewer for bringing up this concern. We believe we have addressed to the best of our ability this issue by including a comprehensive set of questions (4 for each stage), which is likely to be more thorough compared to other studies (including the original paper we have

relied on closely). In addition to these measures, to enhance data quality we have also employed several others as discussed in the "*Inclusion criteria and data quality*" subsection. We agree that future studies would benefit from a combination of lab-based and online samples (both with their respective advantages and disadvantages).

While we indeed documented a relatively high rate of comprehension failures, in respondents' justifications the indication of confusion or misunderstanding over the rules of the game were minimal; less than 4% in any of the experimental conditions with an extremely generous understanding of confusion which includes simple misbelief of the identities of interaction partners.

2) While the use of the term "consensus" is perfectly fine, you may be interested to know that other work in similar cooperative domains has referred to a similar concept as "second-order normative beliefs" (Jachimowicz et al. *Nature Human Behaviour* 2018) or "meta-norms" (Eriksson et al. *Management and Organization Review* 2017).

To enhance the accessibility of our work, we now added these references when discussing our measures.

REVIEWER COMMENTS

Reviewer #1 (Remarks to the Author):

I've read the revision and the reply letter, and appreciate the length the authors went to address the concerns and issues raised by me and the other reviewers.

My main concern with the paper remains. It is unclear whether sharing money with bots is a good abstraction of human-robot interaction, and what meaning participants saw in this sharing.

The authors make a strong argument that bots do not have a use for anything, and therefore there is no disadvantage to using incentivized decision-making tasks to model human-bot interactions. I see the merit in this argument but am not convinced. Two main questions need to be considered:

1. Decision-making with monetary consequences is used in research to model human behavior because of the assumption that people have a preference for money – they want more of it. Because of this assumption, we can use decision-making tasks to model other kinds of human interactions (that do not involve money) where people have preferences over the outcomes. Is this assumption true about bots? Probably not, they have no use for money, unless they were programmed to have a preference for it. The authors argue that this does not matter, as bots have no preference for anything. Even if this is true, this is still not a strong enough argument for using monetary decision-making as a model for human-bot interactions, as it is not a good abstraction of other interactions, there is no advantage to using it. More importantly, it is not necessarily accurate that bots have no preferences about anything. They have preferences over the outcomes they are designed to obtain. For example, a bot that “flags” harmful content on social media has a preference, even if not in the full sense a human does, over the outcome (in this case, what gets censored on social media) – the preference it was programmed for. However, a bot does not have a preference over monetary outcomes, unless it's programmed to have them. I agree with the authors' points that bots' preferences are inherently different than humans' preferences, but they still exist, and because bots do not have a need for money, using a decision-making task with monetary outcomes does not necessarily reflect societies where humans and bots interact.

2. This point is about what bots “objectively” care about, and how suitable economic games themselves are for learning about human-bot societies. Another layer of discussion is how people perceive the preferences of bots, regardless of what bots actually care about. The authors could argue that although bots have no preference or utility for money, participants think they do, and therefore because of how participants perceive these interactions, they are meaningful models of human-bot interaction. However, this aspect was not explored at all. For this to be the main contribution of the paper, much more work needs to happen to show 1) what people think the bots' preferences are for money; and 2) that this models at least some sort of “real” human-bot interaction. The current set of studies is not set to answer these questions.

The paper does provide further support for known effects in behavioral science, such that people are sensitive to norms and are concerned by how their actions will be perceived by others. However, unfortunately, I do not think these merit an independent publication in Nature Communications. I'll add that I find the theoretical question – how do people interact with bots in human-bots societies – interesting, important, and timely. This is definitely a worthwhile endeavor.

A few minor points:

The “qualitative” data that starts on line 126 reports statistics (% of participants who mentioned various reasons for their decisions), but does not report whether the difference is statistically significant. It would be useful to readers to know whether the difference between 20% and 24% is significant and whether the difference between 5% and 10% is significant.

The proper test for categorical variables is usually considered to be a non-parametric test, such as a chi-square test. I might be wrong but it might be a more accurate way to test your data than a t-test (e.g., lines 135 and after).

Reviewer #2 (Remarks to the Author):

This manuscript is much improved over the previous version. The presentation of Studies 2 & 3 is much clearer, and Study 4 is a big addition. The qualitative analyses also help to clarify what participants thought about interacting with bots. Most of my concerns have been addressed. I think this paper is acceptable with the minor revisions listed below. The first two are little more substantial, whereas the others are very easy.

1) The results of Studies 3 & 4 are not as strong as the authors present them in main text. For example, for Study 3, the authors present the significant effect of consensus info on trust towards bots who share with humans, but not the non-effect of consensus info on trust towards people who share with bots. The results are reversed in Study 4: there is an effect on trust towards people who share with bots, which the authors report, but there is no effect on trust towards bots who share with humans, and the authors don't report this null effect. These must all be discussed in the main text so that the authors are not selectively presenting the results that match their hypotheses.

2) Also, the meta-analysis combines Studies 3 & 4 and finds that the confidence interval on the effect sizes overlaps with zero, which reduces our confidence in the effects of being told the norms. As such, the authors should tone down their conclusions accordingly about the effects of seeing a consensus.

Small points:

Figure 2 should explain whether the error bars are SE, SD, or 95% CI.

Figure 2: what about the contrasts between punishment other than just the all-human comparison?

Figure 2 seems to suggest that it will present the results of Jordan et al, but doesn't. Is the grey area meant to describe what Jordan et al did? It ends up being confusing, because it makes it sound like these were Jordan et al's results. I would remove that box because it doesn't refer to anything in the Figure. Note: If the all-human results were from Jordan et al and not from the present study, then this is problematic because a few years have elapsed and the participant pool is likely very different from when Jordan et al ran their study. This needs to be clarified in a response – I'm assuming that the all-human data are from this study, not from Jordan et al, and have made my reviewing decision based on that assumption. Please note that I would seriously reconsider my "minor revisions" if the all-human results were simply drawn from Jordan et al – the authors should clarify to the editor to ensure that the results in that condition are original and not just the data from Jordan et al.

Same as above with Figures S1-S3 – remove the grey part referring to Jordan et al., because it makes it sounds like these data are from their study (as opposed to just replicating their study).

Tables 1 & 2 should mention in the title what experiment they refer to. Also, instead of check marks for the inclusion of those variables, the authors should present the actual coefficients. Same with Tables S1-S4.

Figure S1: H3b-2 should presumably be "... for not sharing with a bot rather than NOT sharing with a person" (emphasis on the missing word)

Figure S10: needs more description of what is going on, and who is making the decisions in the bars. If this graph purports to show that sharing & punishment are costly signals of trustworthiness, then presumably the bars are the proportions returned (i.e., repaid trust) by Player 1s who did/didn't share and Player 3s who did/didn't punish. If so, describe that. Have something in the legend that the darker bars are participants who did share/punish and the lighter bars are participants who didn't share/punish.

Also in Figure S10, shouldn't the asterisks (i.e., who the participant is) be on the blue player

(Experiment 1) or the green player (Experiment 2), instead of the purple player? If this is about trustworthiness of sharers/non-sharers (Player 1), then it's the sharers/non-sharers who are making the decisions, isn't it? Not the purple Player 4s? Similarly, the bars should be coloured blue and green for Experiments 1 & 2, respectively, not purple. Unless I've completely misunderstood what this figure is about, in which case it definitely needs to be clarified.

Reviewer #3 (Remarks to the Author):

The authors have addressed all my concerns and I appreciate the addition of the new study. I don't have any other concerns that need to be addressed, and look forward to seeing the paper published.

We are grateful to the Editor and Reviewers for considering our manuscript, and for their careful attention to the changes we have made. Below we first respond to the direction outlined for us by the Editor who encouraged us to consider the feedback we received from Reviewer 1 and Reviewer 2 on two crucial points.

We begin by assuaging the concern of Reviewer 2. All the data presented in the figures are from our own experiments; we did not extract any data from Jordan et al. (2016) which inspired our research design. We can see how the misunderstanding happened, and we corrected the figures to avoid it.

We also fielded a new study with 299 participants to address the concerns of Reviewer 1. We believe that this study strongly demonstrates that sharing money with bots in our game is an adequate abstraction of real interactions in human-machine collectives such as Twitter and Wikipedia. The study and its findings are described in more detail below. This study is now described as “Study 1” in the revised manuscript, and the other studies are now labeled as Study 2-5. In the Methods section we make clear the chronological order and reference the studies conducted as part of the review process.

We now turn to addressing the comments of the Reviewers point-by-point.

Reviewer #1 (Remarks to the Author):

I’ve read the revision and the reply letter, and appreciate the length the authors went to address the concerns and issues raised by me and the other reviewers.

My main concern with the paper remains. It is unclear whether sharing money with bots is a good abstraction of human-robot interaction, and what meaning participants saw in this sharing. The authors make a strong argument that bots do not have a use for anything, and therefore there is no disadvantage to using incentivized decision-making tasks to model human-bot interactions. I see the merit in this argument but am not convinced. Two main questions need to be considered:

1. Decision-making with monetary consequences is used in research to model human behavior because of the assumption that people have a preference for money – they want more of it. Because of this assumption, we can use decision-making tasks to model other kinds of human interactions (that do not involve money) where people have preferences over the outcomes. Is this assumption true about bots? Probably not, they have no use for money, unless they were programmed to have a preference for it. The authors argue that this does not matter, as bots have no preference for anything. Even if this is true, this is still not a strong enough argument for using monetary decision-making as a model for human-bot interactions, as it is not a good abstraction of other interactions, there is no advantage to using it. More importantly, it is not necessarily accurate that bots have no preferences about anything. They have preferences over the outcomes they are designed to obtain. For example, a bot that “flags” harmful content on social media has a preference, even if not in the full sense a human does, over the outcome (in this case, what gets censored on social media) – the preference it was programmed for. However, a bot does not have a preference over monetary outcomes, unless it’s programmed to

have them. I agree with the authors' points that bots' preferences are inherently different than humans' preferences, but they still exist, and because bots do not have a need for money, using a decision-making task with monetary outcomes does not necessarily reflect societies where humans and bots interact.

2. This point is about what bots “objectively” care about, and how suitable economic games themselves are for learning about human-bot societies. Another layer of discussion is how people perceive the preferences of bots, regardless of what bots actually care about. The authors could argue that although bots have no preference or utility for money, participants think they do, and therefore because of how participants perceive these interactions, they are meaningful models of human-bot interaction. However, this aspect was not explored at all. For this to be the main contribution of the paper, much more work needs to happen to show 1) what people think the bots' preferences are for money; and 2) that this models at least some sort of “real” human-bot interaction. The current set of studies is not set to answer these questions.

We thank the Reviewer for elaborating on their concern and how to address it. These explanations were extremely helpful and guided our design of an appropriate new study (appearing as Study 1 in the revised manuscript). The Reviewer helpfully points out two meanings of “having a preference” for something. Bots can have a preference for money or other outcomes because they feel a need or desire for it as humans do, or they can have a preference for money or other outcomes in the sense that they are programmed to behave as if they wanted them. The question is then whether people think that bots have preferences in the first or second sense, and if so, whether money is a good model for the real-life currencies of human-bot interactions (such as likes on social media, etc.).

To answer these questions, we explained our stylized society to 299 participants and asked them if humans wanted money in this society in the sense of having a need or desire for it (participants on average strongly agreed), if bots wanted money in this society in the sense of having a need or desire for it (participants on average strongly disagreed), and if bots wanted money in this society in the sense of behaving as if they had this preference (participants on average agreed).

Then, importantly, we asked these three questions with respect to two real-life currencies of online human-bot interactions: collecting Likes on Twitter, and avoiding bans on Wikipedia. Participants' responses about these two real-life contexts were remarkably similar to their responses about our stylized society. We believe that these very clear results address the Reviewer's concern, and lend adequate external validity to our other results. We are sincerely grateful to the Reviewer for pushing us on these issues, since we believe that these new findings will be very helpful to other researchers who might also need to justify their choice of using economic games as a proxy for real-life human-bot interactions.

These changes are now discussed in multiple places in the manuscript:

- The Introduction, where earlier we made this case as a form of an assumption.
- The Result section, where we discuss the newly added Fig. 2, pasted below.

- The Methods section, where we elaborate on the new study and highlight its differences from the ones we previously conducted.

Figure 2: Results of Study 1.

The paper does provide further support for known effects in behavioral science, such that people are sensitive to norms and are concerned by how their actions will be perceived by others. However, unfortunately, I do not think these merit an independent publication in Nature Communications. I'll add that I find the theoretical question – how do people interact with bots in human-bots societies – interesting, important, and timely. This is definitely a worthwhile endeavor.

We are pleased that the Reviewer finds our theoretical question important and timely. We also agree that it is reassuring to see the results aligning with our theoretical expectations. Having said that, we respectfully disagree that the effects we find were known. Showing that people can be influenced by social norms and reputation into being prosocial to *machines* is a novel and exciting contribution of our manuscript.

A few minor points:

The “qualitative” data that starts on line 126 reports statistics (% of participants who mentioned various reasons for their decisions), but does not report whether the difference is statistically significant. It would be useful to readers to know whether the difference between 20% and 24% is significant and whether the difference between 5% and 10% is significant. The proper test for categorical variables is usually considered to be a non-parametric test, such as a chi-square test. I might be wrong but it might be a more accurate way to test your data than a t-test (e.g., lines 135 and after).

We thank the Reviewer for this suggestion. We have now included p-values using McNemar's ratio test in our discussion in the main text. However, it is important to note that we consider the qualitative data as mostly useful for descriptive rather than inferential purposes. Future research might form categories explicitly asked about in survey instruments, to get an even more comprehensive picture of the reasons people give for making decisions in the incentivized game we study, and we believe that formal hypothesis tests are more appropriate in that case.

Reviewer #2 (Remarks to the Author):

This manuscript is much improved over the previous version. The presentation of Studies 2 & 3 is much clearer, and Study 4 is a big addition. The qualitative analyses also help to clarify what participants thought about interacting with bots. Most of my concerns have been addressed. I think this paper is acceptable with the minor revisions listed below. The first two are little more substantial, whereas the others are very easy.

1) The results of Studies 3 & 4 are not as strong as the authors present them in main text. For example, for Study 3, the authors present the significant effect of consensus info on trust towards bots who share with humans, but not the non-effect of consensus info on trust towards people who share with bots. The results are reversed in Study 4: there is an effect on trust towards people who share with bots, which the authors report, but there is no effect on trust towards bots who share with humans, and the authors don't report this null effect. These must all be discussed in the main text so that the authors are not selectively presenting the results that match their hypotheses.

We thank the Reviewer for their close attention to our paper, and indeed, we agree that we should update our discussion where we contrast the results between Studies 3 and 4. We now write in the Results section:

“Results from Studies 4 and 5 complement one another, see SM Note 11, for an extensive meta-analysis. While the results from these two studies are different (one finds an effect in one condition, but not in the other and the findings in the second study are reversed), taken together, these studies show that our data are consistent with generally positive effects in case of bot Helpers (while the respective CI contains 0, leaving the possibility for a null-effect), and positive effects in case of human Helpers interacting with bot Beneficiaries.”

We have also updated our discussion of the meta-analysis in the Methods section to be more cautious in our interpretation, and also to further highlight the differences in findings for these two studies. In particular, we now write the following (please note that Study 4 and 5 were earlier Study 3 and 4):

“We dedicate a supplementary note (Note 11) to meta-analyze Study 4 and Study 5 following Morris and DeShon [31], as their conclusions, analyzed separately are not

identical (see Fig. 4). The upshot of this analysis is twofold: we find weak evidence for the homogeneity assumption, namely that the treatment effects are the same across the between- and within-person design, which is not surprising, as these are two different theoretical quantities. When Helpers are bots, we find an increase in their trust-gain estimate with large variance, falling in the [-0.075, 0.387] 90% window, which is consistent with a large positive effect, but does not rule out a null-effect. When people help bot Beneficiaries, they benefit from the norm-signal as suggested by the [0.042, 0.248] 90% window of the trust-gain. Note however, that these combined results should be interpreted with caution due to lack of evidence that the homogeneity assumption holds.”

2) Also, the meta-analysis combines Studies 3 & 4 and finds that the confidence interval on the effect sizes overlaps with zero, which reduces our confidence in the effects of being told the norms. As such, the authors should tone down their conclusions accordingly about the effects of seeing a consensus.

This is a related point to the one above. We have made the following changes to the discussion section to address these concerns in the Discussion section at two points. For convenience, we quote these sentences below:

“In addition, we follow up by demonstrating that trust-gains generally increase when informing participants of the high consensus about the norm of sharing, providing complementary evidence for this across two studies with different designs, suggesting that people might alter their behavior once informed.”

“We show in particular that a consensus about prosocial norms can emerge in these collectives, faster than people might expect---and that informing them about this emerging consensus could accelerate the pace at which trust is established within the collective.”

Small points:

Figure 2 should explain whether the error bars are SE, SD, or 95% CI.

The caption for the figure has been updated to reflect that the error bars are the standard errors of the respective means.

Figure 2: what about the contrasts between punishment other than just the all-human comparison?

We agree with the Reviewer that further analysis (and theorizing) of the additional comparisons will be a fruitful avenue for future work. As we have considered the baseline as the human only condition, we include the results of the tests we pre-registered that compares the conditions that contain bots to the human only condition.

Figure 2 seems to suggest that it will present the results of Jordan et al, but doesn't. Is the grey area meant to describe what Jordan et al did? It ends up being confusing, because it makes it sound like these were Jordan et al's results. I would remove that box because it doesn't refer to anything in the Figure. Note: If the all-human results were from Jordan et al and not from the present study, then this is problematic because a few years have elapsed and the participant pool is likely very different from when Jordan et al ran their study. This needs to be clarified in a response – I'm assuming that the all-human data are from this study, not from Jordan et al, and have made my reviewing decision based on that assumption. Please note that I would seriously reconsider my "minor revisions" if the all-human results were simply drawn from Jordan et al – the authors should clarify to the editor to ensure that the results in that condition are original and not just the data from Jordan et al.

Thank you for this surgical comment. All data and analysis, including the human only conditions, have been collected by us. We have revised the text and the figures to ensure there is no ambiguity about this, specifically in the first paragraph of the Recruitment subsection in Methods, and Fig. 2. We agree with the Reviewer that the time difference could have raised questions, which is why we have opted for the experimental design where we recruited all respondents whose data we analyzed.

Same as above with Figures S1-S3 – remove the grey part referring to Jordan et al., because it makes it sounds like these data are from their study (as opposed to just replicating their study).

We took this specific suggestion of the Reviewer and removed the grey part from both Fig. 2 (which is now Fig. 3 in the revised manuscript) and the corresponding supplementary figures mentioned above.

Tables 1 & 2 should mention in the title what experiment they refer to. Also, instead of check marks for the inclusion of those variables, the authors should present the actual coefficients. Same with Tables S1-S4.

We thank the Reviewer for these suggestions. We present the complete regression table in the Supplementary Materials (Tables S5 and S6). Placing the table in the main paper would include an additional 11 point estimates, none of which should be considered causal effects. Thus, keeping the table in its current form would have the advantage of reducing clutter and avoiding over-interpretation. We hope the Reviewer agrees. Additionally, we now refer the interested reader to the relevant SI table in the figure caption.

Figure S1: H3b-2 should presumably be "... for not sharing with a bot rather than NOT sharing with a person" (emphasis on the missing word)

We thank the Reviewer for pointing this out. The text in this figure has been corrected.

Figure S10: needs more description of what is going on, and who is making the decisions in the bars. If this graph purports to show that sharing & punishment are costly signals of trustworthiness, then presumably the bars are the proportions returned (i.e., repaid trust) by Player 1s who did/didn't share and Player 3s who did/didn't punish. If so, describe that. Have something in the legend that the darker bars are participants who did share/punish and the lighter bars are participants who didn't share/punish.

We agree with the Reviewer that the explanation of the bars should be clarified. Hence, we updated the caption to reflect that:

“The bars quantify the average proportion returned by Players 1 and 3 in the trust game. The darker colored bars correspond to participants who shared or punished in the first stage, while lighter ones to participants who didn't share or didn't punish.”

Also in Figure S10, shouldn't the asterisks (i.e., who the participant is) be on the blue player (Experiment 1) or the green player (Experiment 2), instead of the purple player? If this is about trustworthiness of sharers/non-sharers (Player 1), then it's the sharers/non-sharers who are making the decisions, isn't it? Not the purple Player 4s? Similarly, the bars should be coloured blue and green for Experiments 1 & 2, respectively, not purple. Unless I've completely misunderstood what this figure is about, in which case it definitely needs to be clarified.

We thank the Reviewer for their astute comment. Indeed, the bars refer to the behavior of Players 1 and Players 3, respectively, and therefore, it is appropriate to signal these in blue and green colors.

Reviewer #3 (Remarks to the Author):

The authors have addressed all my concerns and I appreciate the addition of the new study. I don't have any other concerns that need to be addressed, and look forward to seeing the paper published.

We are grateful to the Reviewer for their time and consideration of our work.

REVIEWER COMMENTS

Reviewer #1 (Remarks to the Author):

The authors have fully addressed my concerns by adding the new study and discussing the assumptions in the introduction.

Just a small note -- in the discussion you still mention only 4 studies. I wasn't sure if that was intentional or not.

Reviewer #2 (Remarks to the Author):

I thank the authors for making their revisions, and in particular for collecting more data on how people perceive bots' "desire" for money. I'm actually a bit less warm on the manuscript after seeing those results: most participants acknowledge that bots don't really want money (or anything else), which suggests that money might not be a good currency for such an experiment. Yes, participants said that bots act "as if" they wanted those currencies, but I'm not sure what that says. After all, people might say that calculators act "as if they wanted to calculate", or automatic doors act "as if they want to open when people come near". So I'm not sure exactly what it means if participants say that bots act "as if they want money". The authors would need to justify this more. The authors might be able to parlay that into a strength: people are still giving something to bots, even though bots don't really want the money, simply because they want to gain reputation for giving (or avoid a bad reputation for not-giving).

I just noticed that for many/most comparisons, the authors present just the p-values, not the effect size. Whenever possible, they should present an effect size in a standard currency (e.g., Cohen's d, Hodges g) so that readers can interpret how big an effect it is. This is a minor change.

In study 1, the authors should give the stats for pairwise comparisons between currencies. This can be in supplementary to save space in the main text.

Line 183 should read "study 2"

Dear Editors and Reviewers,

Thank you for your time and attention to our work. We apologize for the time it took to turn around our manuscript for this last round of revisions. During the summer our holiday schedules were non-overlapping, all authors traveled, some relocated, which made it challenging to execute the final strokes to the paper.

Below we address the comments of Reviewer 1 and Reviewer 2.

Reviewer #1 (Remarks to the Author):

The authors have fully addressed my concerns by adding the new study and discussing the assumptions in the introduction.

Just a small note -- in the discussion you still mention only 4 studies. I wasn't sure if that was intentional or not.

Thank you for pointing this out, it was an oversight. The Editors also drew our attention to this inconsistency in two places, which we have now corrected.

Reviewer #2 (Remarks to the Author):

I thank the authors for making their revisions, and in particular for collecting more data on how people perceive bots' "desire" for money. I'm actually a bit less warm on the manuscript after seeing those results: most participants acknowledge that bots don't really want money (or anything else), which suggests that money might not be a good currency for such an experiment. Yes, participants said that bots act "as if" they wanted those currencies, but I'm not sure what that says. After all, people might say that calculators act "as if they wanted to calculate", or automatic doors act "as if they want to open when people come near". So I'm not sure exactly what it means if participants say that bots act "as if they want money". The authors would need to justify this more. The authors might be able to parlay that into a strength: people are still giving something to bots, even though bots don't really want the money, simply because they want to gain reputation for giving (or avoid a bad reputation for not-giving).

We have very extensively addressed these questions in the revised Introduction and in the new Study 1. We have now added a paragraph in the discussion section. For context, this is what we write in the Introduction:

"Introducing bots in our stylized society raises a methodological and a theoretical question. From a methodological perspective, the issue at stake is that bots, unlike humans, do not care about money. As a result, one may question our choice to use economic games as a proxy for real-life interactions between bots and humans. To justify our paradigm, we show that human participants believe that bot participants behave as if they had preferences, be it for money (the currency in our experiment), or for real-life currencies like collecting likes on social media, or avoiding sanction and bans in online communities such as Wikipedia. This is what we accomplish in Study 1. More precisely, we demonstrate that while human participants do not believe that bots 'want' to earn money, collect likes, or avoid bans (in the sense that they feel a need or desire for these

outcomes), they do believe that the bots behave as if they had all these preferences, because of the way they are programmed. This being established, the theoretical question is why people may share resources with bots, money or otherwise. While altruism is an unlikely explanation [25], confusion is always an option [26], although we take extensive precautions so that participants understand the incentive structure of our stylized society. The explanation we focus on, instead, is signaling: humans help bots in order to signal to other humans that they are trustworthy.”

We have added the following paragraph in the Discussion, revisiting this question and incorporating this new comment:

“It is generally accepted that stylized societies featuring incentivized interactions are an adequate way to capture the many other ways humans cooperate outside the lab. In a nutshell, humans want money, so money is a good currency for cooperation studies in the lab. Humans and bots cooperate in many ways outside the lab, too, but it is less clear that money is a good currency for human-bot cooperation studies in the lab, since bots have no use for money. Indeed, our participants said that bots had no desire for money, although they also said that bots acted as if they wanted money, or retweets. In other words, humans may have behaved in our stylized society in ways that are consistent with their expectations that bots would be have according to wanting money. While this is a possibility, the evidence presented here is consistent with people retweeting bots, or giving them money in stylized societies because prosociality toward bots is a good signal to send to other humans.”

In addition, we also reflect on this in the Methods:

“Although sharing may seem to be motivated purely by altruism, Jordan and colleagues provide evidence that it may also serve a strategic purpose rooted in the incentives that are experienced in daily life and are made explicit in the game when interacting with Trustors [24] (see also earlier work on the reputational benefits, i.e., the signaling value of punishment [43, 44]). Our interpretation of sharing is consistent with theirs, and is confirmed by analyzing the reasons that participants give when making their helping decisions (for additional details, see SM Note 12).”

I just noticed that for many/most comparisons, the authors present just the p-values, not the effect size. Whenever possible, they should present an effect size in a standard currency (e.g., Cohen’s *d*, Hodges *g*) so that readers can interpret how big an effect it is. This is a minor change.

Thank you for raising this point. With the Editors’ guidance we addressed all places with additional detail for the statistical analysis we conduct. We also reviewed the paper to include sizes where missing using the key quantities we defined.

In study 1, the authors should give the stats for pairwise comparisons between currencies. This can be in supplementary to save space in the main text.

We have now added more language to the main text, mentioning means. We added all detail necessary to the statistical tests we conduct but establishing numerical equality among these quantities has not been our goal.

Line 183 should read “study 2”

Thank you for catching this. We believe this may have been a misunderstanding given the introduction of the new study, which took the label “Study 1.” We have read the manuscript carefully and made sure to correct any typos in the numbering.